# Acetyl-CoA synthetase activity is enzymatically regulated by lysine acetylation using acetyl-CoA or acetyl-phosphate as donor molecule

Chuan Qin[1], Leonie G. Graf[1], Kilian Striska[1], Markus Janetzky[1], Norman Geist [2], Robin Specht [3], Sabrina Schulze [1], Gottfried J. Palm [1], Britta Girbardt[1], Babett Dörre[1], Leona Berndt[1], Stefan Kemnitz[4], Mark Doerr [3], Uwe T. Bornscheuer [3], Mihaela Delcea [2] & Michael Lammers [1] ✉

The AMP-forming acetyl-CoA synthetase is regulated by lysine acetylation both in bacteria and eukaryotes. However, the underlying mechanism is poorly understood. The *Bacillus subtilis* acetyltransferase AcuA and the AMP-forming acetyl-CoA synthetase AcsA form an AcuA•AcsA complex, dissociating upon lysine acetylation of AcsA by AcuA. Crystal structures of AcsA from *Chloroflexota bacterium* in the apo form and in complex with acetyl-adenosine-5′-monophosphate (acetyl-AMP) support the flexible C-terminal domain adopting different conformations. AlphaFold2 predictions suggest binding of AcuA stabilizes AcsA in an undescribed conformation. We show the AcuA•AcsA complex dissociates upon acetyl-coenzyme A (acetyl-CoA) dependent acetylation of AcsA by AcuA. We discover an intrinsic phosphotransacetylase activity enabling AcuA•AcsA generating acetyl-CoA from acetyl-phosphate (AcP) and coenzyme A (CoA) used by AcuA to acetylate and inactivate AcsA. Here, we provide mechanistic insights into the regulation of AMP-forming acetyl-CoA synthetases by lysine acetylation and discover an intrinsic phosphotransacetylase allowing modulation of its activity based on AcP and CoA levels.

Lysine acetylation constitutes an important post-translational modification in all domains of life[1,2]. Lysine acetylation is extensively studied in eukaryotes, and also in bacteria it is known that lysine acetylation is used to sense the cellular metabolic state and translates this directly into altered protein functionalities allowing it to adjust to rapidly changing environmental conditions[2–4]. In bacteria N-(ε)-lysine acetylation is catalysed enzymatically by Gcn5-related N-terminal acetyltransferases (GNATs) using acetyl-CoA as a donor molecule for lysine acetylation[5–8]. These enzymes structurally resemble mammalian GNATs in their active site architecture[5,7,9–12]. Bacterial GNATs are classified into several different subgroups depending on the presence of additional C- and/or N-terminal domains, which were shown to either interfere with folding of the catalytic domain, to allosterically modulate GNAT catalytic activity upon ligand binding, which affects

[1]Department of Synthetic and Structural Biochemistry, Institute of Biochemistry, University of Greifswald, 17489 Greifswald, Germany. [2]Department of Biophysical Chemistry, Institute of Biochemistry, University of Greifswald, 17489 Greifswald, Germany. [3]Department of Biotechnology & Enzyme Catalysis, Institute of Biochemistry, University of Greifswald, 17489 Greifswald, Germany. [4]Department for High Performance Computing, University Computing Center, University of Greifswald, 17489 Greifswald, Germany. ✉e-mail: michael.lammers@uni-greifswald.de

oligomerization, or to be of yet unknown physiological function[12–26]. Besides lysine side chains, some GNATs are reported to possess N-(α)-acetyltransferase activity, i.e. enabling the acetylation of α-amino groups of proteins. Moreover, conflicting data exist in the literature reporting that some GNATs act as dual N-(α)- and N-(ε)-acetyltransferases[4,6,10,11,13,22,23]. The deacetylation of acetylated lysine side chains is catalysed by classical $Zn^{2+}$-dependent lysine deacetylases and by $NAD^+$-dependent sirtuin deacetylases[27–36]. Members of these classes differ structurally and apply different catalytic strategies for deacetylation. While bacterial sirtuins are structurally and functionally well characterised much less is known about bacterial $Zn^{2+}$-dependent classical deacetylases[27,37–41].

For *B. subtilis* it was shown that AMP-forming acetyl-CoA (coenzyme A)-synthetase (AcsA) activity is regulated by acetylation of a lysine side chain (K549) in core region A10 of the C-terminal domain[34,36,42]. This lysine is conserved in AMP-forming acetyl-CoA synthetases and other acyl-AMP-forming enzymes in bacteria, archaea and eukaryotes suggesting that the regulation by lysine acetylation is evolutionary conserved[35,43–56]. Mutation of the homologously conserved lysine strongly impairs acyl-adenylation activity in several cases as it is important for nucleotide binding and to position the carboxylate substrate for nucleophilic attack on the α-phosphate of ATP[52,57]. This class of enzymes encompasses besides the AMP-forming acetyl-CoA synthetases also the acyl-/aryl-CoA-synthetases/ligases as well as adenylation domains of non-ribosomal peptide synthetases (NRPSs) and firefly-luciferases, which are classified as ANL-superfamily[58]. All members share app. 20% sequence identity[48,58–72]. Subsequently, other enzymes including fatty acyl-AMP ligases, aryl polyene adenylation enzymes, and β-lactone synthetases were also classified into this ANL-superfamily[62,73–75]. All these enzymes share the mechanistic step of generating acyl-AMP during catalysis. Despite exhibiting this common adenylation step, different enzymes of the superfamily catalyse different overall reactions comprising an example of divergent evolution[58]. Structurally, AMP-forming acetyl-CoA synthetases contain a large N-terminal domain of app. 450-500 amino acids composed of two mostly parallel eight-stranded β-sheets, i.e. β-sheet A and β-sheet B, surrounded by several α-helices providing binding sites for CoA and ATP, and an additional β-sheet consisting of four antiparallel β-strands forming a β-barrel (β-sheet C)[51,70,72]. These β-sheets are flanked by several α-helices (Supplementary Fig. 13a)[51,70,72]. The smaller C-terminal domain of app. 110–130 amino acids is composed of a small β-hairpin consisting of a two-stranded antiparallel β-sheet and a three-stranded β-sheet surrounded by two α-helices on each side[70,72]. Several conserved sequence motifs contain important residues for catalysis as well as CoA and nucleotide binding[51], which are classified into either core sequences A1 to A10 or into motifs I, II, and III (Supplementary Fig. 1)[69,71]. These include a Ser-/Gly-rich loop (core region A3/motif III; phosphate binding loop; orientation of β,γ-phosphates of ATP; binding of the leaving group pyrophosphate), a sequence containing a Thr-Glu dipeptide sequence (AcsA: DTWWQTET; core region A5/motif II; first aromatic side chain stacks to adenine ring of ATP, Thr contacts phosphate of acyl-AMP; Glu coordinates $Mg^{2+}$), core region A7/motif III with the consensus sequence (S/T)GD (conserved aspartate binds hydroxyls of ATP-ribose), an RX(D/K)X$_6$G sequence (core region A8; Arg contacts hydroxyls of acyl-AMP-ribose, Asp/Lys form a hinge, Gly lines CoA 4-phosphopantetheine tunnel in thioester conformation), and a region containing a conserved Gly-Lys dipeptide (core region A10; lysine is in the active site in adenylation-conformation) (Supplementary Figs. 1, 6, 11, and 12)[58,70,71]. Most bacterial AMP-forming acetyl-CoA synthetases were reported to stay as monomers or dimers in solution, while the yeast enzyme is trimeric[51,63,68,76–78]. AMP-forming acetyl-CoA synthetases catalyse the ATP-dependent formation of acetyl-CoA from acetate and coenzyme A in two half-reactions (Fig. 1a)[54]. In the first half-reaction, acetyl-CoA synthetase activates acetate using ATP to form acetyl-AMP under the release of

pyrophosphate PP$_i$ (adenylation reaction). In the second half-reaction, the thioester acetyl-CoA is generated from the mixed anhydride acetyl-AMP and CoA releasing AMP (thioesterification/thioester-formation) (Fig. 1a)[54]. To enable both half-reactions during catalysis, acyl-AMP-forming enzymes adopt several conformations allowing binding of the required co-substrates and arranging the catalytic machinery[21,35,51,53,58]. These conformations differ only in the orientation of the C-terminal domain relative to the N-terminal domain, while the integrity of the N-terminal domains stays almost identical[51,54,58,63,68,70,72,79]. Overall, three conformations were reported, i.e. the apo form described for *Photinus pyralis* firefly luciferase (PDB: 1LCI; open conformation; Supplementary Fig. 12a), the conformation to catalyse the first half-reaction (adenylation-conformation) reported for *Brevibacillus brevis* phenylalanine activating domain of gramicidin synthetase 1 (PDB: 1AMU; Supplementary Fig. 12b) and the conformation to catalyse the second half-reaction (general: AMP-release; in AcsA: thioester-forming conformation) reported for *Salmonella enterica* AMP-forming acetyl-CoA synthetase Acs (PDB: 1PG3, 1PG4; Supplementary Fig. 12c, d)[51,70,72].

*B. subtilis* AcsA is reversibly transcribed upstream from the polycistronic *acu*-operon encompassing the genes *acuA*, *acuB* and *acuC* (*acu*: acetoin utilization; Fig. 1a). Initially, the *acuABC* gene products were identified enrolling in the metabolism and degradation of acetoin and butanediol during growth and sporulation[34,42,80,81]. Expression of the *acu*-operon was shown to be high in the stationary phase and repressed upon the addition of glucose as a carbon source[42]. AcuA was described as a GNAT-related lysine acetyltransferase (KAT) catalysing the acetyl-CoA-dependent K549-acetylation thereby inhibiting AcsA activity (Fig. 1a)[34,36,80]. AcuC is a $Zn^{2+}$-dependent classical deacetylase capable to deacetylate AcsA at K549 thereby (re-)activating AcsA (Fig. 1)[36]. It is not known whether AcuC or AcuA have further substrates besides AcsA. The protein AcuB is of unknown function. It contains two predicted cystathionine-β-synthase (CBS)-domains in the N-terminus and a C-terminal ACT-domain (ACT: aspartate kinase, chorismate mutase, TyrA). Some bacterial GNAT KATs of type III contain an N-terminal ACT domain preceding the catalytic KAT domain[2,3,82,83]. The binding of amino acids cysteine, arginine or asparagine to these ACT domains was shown to activate the KAT activity[3,82,84–86]. CBS-domains were described to bind adenine nucleotides and $NAD^+$[87–90]. It is not known whether AcuB plays a similar role in regulating AcuA or AcuC activity upon binding to specific metabolites. Acetylation of AcsA catalysed by the acetyl-CoA dependent lysine acetyltransferase AcuA constitutes a regulatory lysine acetylation-mediated feedback loop controlling the acetyl-CoA level[34,36,80,91]. Next to the enzymatically catalysed acetylation of lysine side chains, also non-enzymatic acetylation has been described in eukaryotes and prokaryotes[38,92–95]. This non-enzymatic acetylation depends on various factors such as cellular acetyl-CoA levels and the intracellular pH value[38,96]. Moreover, as the efficiency of non-enzymatic acetylation also depends on the amino acid composition a site-specificity of non-enzymatic acetylation has been proposed[96]. In prokaryotes, acetyl-phosphate (AcP) has been identified as the major source of non-enzymatic lysine acetylation[97,98]. AcP is produced in bacteria under conditions of carbon overflow metabolism by the activities of acetate kinase (AK), generating AcP from acetate and ATP (acetate assimilation), or by phosphotransacetylase (Pta) using acetyl-CoA and phosphate[47,48,99]. As the AK/Pta pathway is reversible, in contrast to the reactions catalysed by AcsA, it is important for generating acetate when intracellular concentrations for acetyl-CoA and ATP are high (acetate dissimilation)[47,48,99,100]. Thereby, the AK/Pta pathway couples energy metabolism to the metabolism of carbon and phosphorus[100,101]. The biochemical properties of the enzymes AcsA and AK/Pta and the cellular concentrations of acetate, AcP and acetyl-CoA determine which reactions are predominant under physiological conditions. AK has a $K_M$-value for acetate in the millimolar range suggesting activity of the AK/Pta pathway for acetate assimilation only at very high intracellular

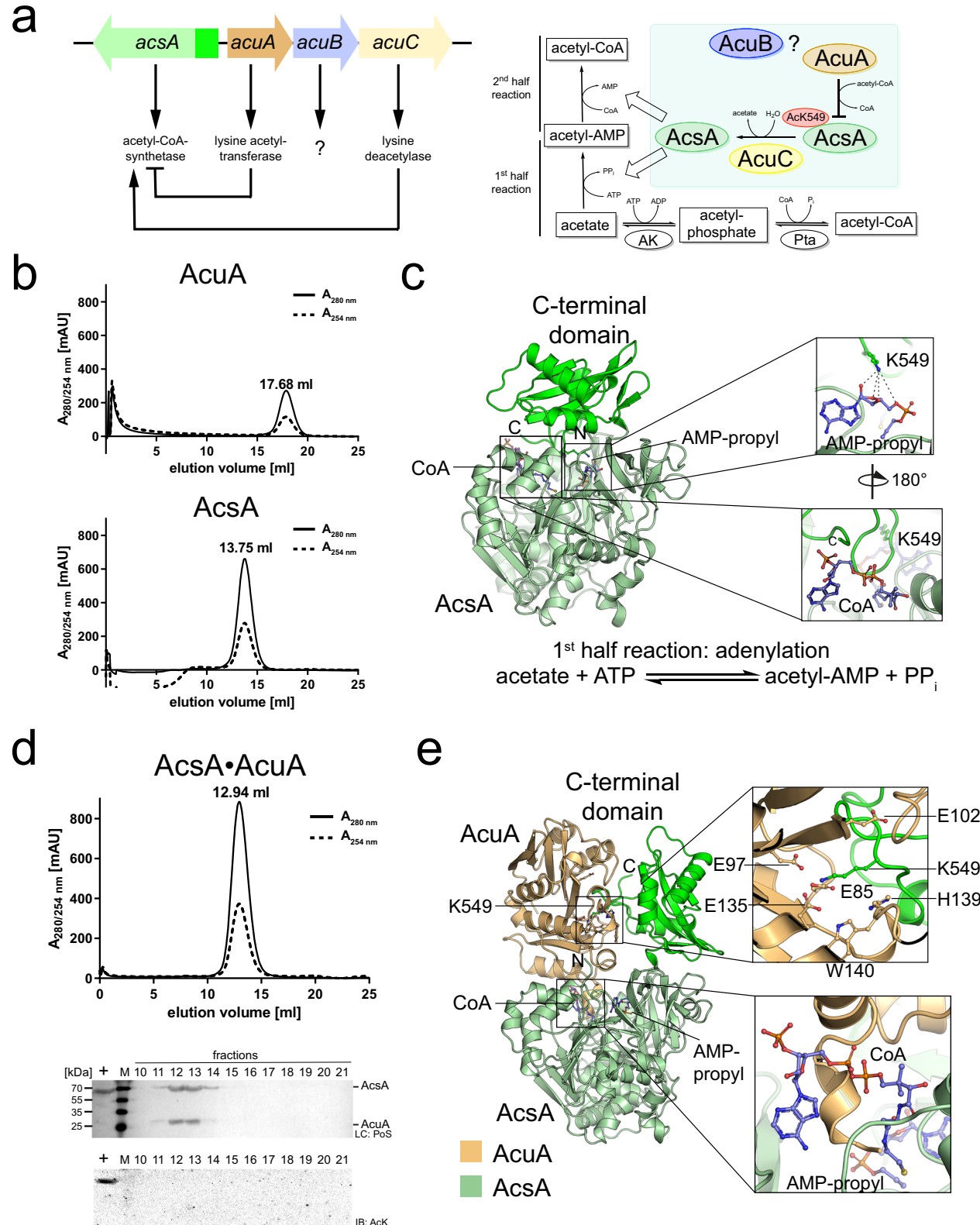

concentrations of acetate[102]. So far, no example has been reported for an enzymatically catalysed lysine acetylation using acetyl-phosphate as an acetyl-group donor molecule.

In this work, we provide detailed mechanistic data on the regulation of AcsA activity by lysine acetylation and report an example of an enzymatically catalysed protein lysine acetylation using AcP as an acetyl-group donor by coupling Pta activity with lysine acetyltransferase activity.

## Results

### *B. subtilis* AcsA forms a stable complex with AcuA

In *B. subtilis* the proteins AcuA, AcuB and AcuC are encoded in the *acu*-operon and AcsA is reversely transcribed upstream of the operon (Fig. 1a). We obtained highly pure proteins for AcsA, AcuA, AcuB and AcuC in a yield suitable for biochemical studies (Supplementary Fig. 2a). Analytical size exclusion chromatography (SEC) revealed for AcsA a retention volume corresponding to a globular dimer (exp. MW:

**Fig. 1 | *Bacillus subtilis* AcsA and AcuA form a stable complex. a** The *B. subtilis acu*-operon encodes for the lysine acetyltransferase AcuA, the classical $Zn^{2+}$-dependent KDAC AcuC and AcuB of unknown function. The acetyl-CoA synthetase AcsA is reversely transcribed upstream of the *acu*-operon. Acetylation of K549 in the AcsA C-terminal domain (dark green) inhibits AcsA activity, deacetylation by AcuC restores its activity. AcsA catalyses the generation of acetyl-CoA in two half-reactions. In the first half-reaction, acetate is converted to acetyl-AMP under consumption of ATP and release of pyrophosphate ($PP_i$), making the reaction irreversible (adenylation reaction) and in the second half-reaction the mixed anhydride acetyl-AMP is converted to the thioester acetyl-CoA (thioester-forming reaction). **b** AcuA forms a monomer and AcsA a dimer in solution as shown by analytical size exclusion chromatography (SEC) experiments. Source data are provided as Source Data file. **c** AlphaFold2 structure prediction of AcsA. AcsA is in the conformation to catalyse the first half-reaction (adenylation reaction). Superimposing the Alpha-Fold2 AcsA structure and AcsA•AMP-propyl•CoA of *S. enterica* (PDB: 1PG3) shows this conformation being incompatible with CoA binding (lower closeup). K549 of AcsA is in direct interaction distance to the ribose of AMP-propyl using the same binding site as ATP (upper closeup). **d** AcsA and AcuA form a stable complex in solution. Analytical SEC experiments suggest that AcsA and AcuA form an apparent heterotetramer composed of two AcsA•AcuA heterodimers. The lane labelled with + shows the acetylated *B. subtilis* AcsA loaded as technical control for the immunoblot, and the lane labelled with M represents the protein molecular weight marker. The experiment was repeated independently three times with similar results. Source data are provided as Source Data file. **e** AcuA binding stabilizes AcsA in a conformation not capable of catalysing the first or second half-reaction. Upon AcuA binding a conformational change of the flexible AcsA C-terminal domain is observed. In this conformation AcsA K549 is placed directly into the active site of AcuA, coordinated by several glutamic/aspartic acid side chains. AcsA is inactive in this conformation as the CoA binding site is occupied by the C-terminal α-helix of AcuA.

66.0 kDa, calc. MW: 147.7 kDa) while AcuA eluted with a retention volume relating to a monomer (exp. MW: 25.5 kDa, calc. MW: 23.7 kDa; Fig. 1b, Supplementary Fig. 2b, c). To generate hypotheses on the structural basis for AcsA dimer formation, we performed AlphaFold2 structure predictions (Fig. 1c; Supplementary Fig. 4a; Supplementary Data 1). The monomeric forms of AcuA or AcsA were predicted with high confidence scores (pLDDT: 94.8 AcuA/92.1 AcsA; Supplementary Fig. 2f; Supplementary Data 1). The prediction of the AcsA•AcsA homodimer was obtained with an ipTM+pTM-confidence score of 0.91 suggesting a high level of accuracy for that AcsA•AcsA model (Supplementary Figs. 2c, f, 3, and 4a; Supplementary Data 1). The calculated dimer interface area is 789.7 Å² (Supplementary Fig. 2d). The predicted AcsA model is similar to the structure of *Brevibacillus brevis* phenylalanine activating domain of gramicidin synthetase 1 in complex with AMP and the substrate phenylalanine (PDB: 1AMU; r.m.s.d. 1.971 Å) suggesting that the predicted AcsA model represents the AcsA in a conformation capable to catalyse the first half-reaction, i.e. adenylation of acetate with ATP to form acetyl-AMP and $PP_i$ (Fig. 1c; Supplementary Figs. 12b and 13c)[70].

Next, we performed analytical size exclusion chromatography (SEC) experiments to analyse whether AcsA and AcuA directly interact. We observe a direct interaction of AcsA and AcuA as they co-elute from the SEC column with a calculated molecular weight of 201 kDa corresponding to a tetramer consisting of two AcsA•AcuA heterodimers (exp. MW: 91 kDa; Fig. 1d; Supplementary Fig. 2c). Analytical SEC experiments performed with equimolar concentrations of AcsA and AcuA suggested that AcsA binds to AcuA with a sub-micromolar affinity as they co-eluted in a single peak (Fig. 1d). Neither AcsA nor AcuA is lysine-acetylated as shown by immunoblotting of the elution fractions of the SEC experiments using a specific anti-acetyl-lysine antibody (anti-AcK-AB) (Fig. 1d). To get more insights into the AcsA•AcuA complex formation as well as into interactions created within the complex we performed more AlphaFold2 structure predictions for the AcsA•AcuA heterodimer (Fig. 1e; Supplementary Fig. 2e; Supplementary Data 1). We obtained a reasonable confidence score (ipTM+pTM: 0.72) indicative for a correct prediction of fold, topology and of the interface (Supplementary Figs. 2f and 3; Supplementary Data 1). These analyses show that within the AcsA•AcuA complex, the C-terminal domain in AcsA performs a conformational change by rotating app. 90° compared to the structure of noncomplexed AcsA (first half-reaction: adenylation-conformation) (Supplementary Fig. 4b). AcuA binds to AcsA by stabilizing AcsA in a conformation that was not reported before. In this predicted conformation the C-terminal domain is neither competent to catalyse the first half-reaction (adenylation reaction) nor to catalyse the second half-reaction (thioester-forming reaction; Fig. 1c, e). According to the AlphaFold2 model, a major factor for arranging the AcsA C-terminal domain upon binding to AcuA into this conformation is placing the K549 of AcsA into the AcuA active site therefore forming electrostatic interactions to D82,

E85, E97, E102, E135 and potentially H139 (Fig. 1e, upper closeup). The calculated interface area of AcuA and AcsA is 1764.6 Å² supporting the experimental data and the formation of a stable complex (Supplementary Fig. 2d). Next, we analysed whether and how the binding of AcuA to AcsA affects the activity of AcsA.

## Binding of AcuA to AcsA abolishes AcsA activity

To analyse whether the binding of AcuA to AcsA affects AcsA activity we assessed the capability of AcsA to synthesize acetyl-CoA from acetate, ATP and CoA with or without preincubation of AcsA with AcuA. The production of acetyl-CoA is detected subsequently by the addition of AcuA assessing AcsA acetylation by immunoblotting using an anti-AcK-AB. This indicates that AcsA alone, i.e. without preincubation with AcuA, is actively utilising both half-reactions resulting in the formation of acetyl-CoA, which is used by AcuA to acetylate AcsA (Fig. 2a). However, preforming a complex between AcuA and AcsA before the addition of ATP, CoA and acetate resulted in a catalytically inactive AcsA enzyme not capable of generating acetyl-CoA as AcuA cannot acetylate AcsA in this case (Fig. 2a). The AlphaFold2 model of the AcsA•AcuA complex allows to postulate a hypothesis for the molecular mechanism underlying the observed inhibitory effect of AcuA upon binding to AcsA (Fig. 1e). Compared to free AcsA, the C-terminal domain of AcsA, within the AcsA•AcuA complex, performs a huge conformational change mostly driven by the formation of electrostatic interactions of AcsA K549 with several negatively charged residues in AcuA (D82, E85, E97, E102, E135; Fig. 1e, upper closeup). K549 is highly conserved within the core motif A10 and is essential for the adenylation reaction as it is needed for nucleotide binding and orientation as well as activation of the substrate (Supplementary Figs. 7 and 13)[70]. To this end, the mere binding of AcsA to AcuA stabilizes AcsA in a conformation not capable of catalysing the first half-reaction (adenylation reaction). The activity of AcsA to catalyse the second half-reaction (thioester-forming reaction) is also compromised in the complex with AcuA as a small C-terminal α-helix of AcuA interferes with CoA binding to AcsA and as several residues (such as motif A8; 464-GER-466; Supplementary Fig. 7) needed for CoA- and acyl-AMP-binding are reoriented (Fig. 1e, lower closeup). Besides from binding of AcuA to AcsA, it was previously reported that acetylation of *B. subtilis* AcsA at K549 in the C-terminal core motif A10 by AcuA results in inhibition of AcsA activity[34,36]. These findings tempted us to investigate how AcsA acetylation affects the integrity of the AcsA•AcuA complex, how acetylation finally results in the inhibition of AcsA activity and why the cell employs two mechanisms to inhibit AcsA activity.

## Acetylation of AcsA dissociates the AcsA•AcuA complex

As we show above binding of AcuA to AcsA inhibits AcsA activity. We next analysed how acetylation of AcsA by AcuA affects the interaction of AcsA and AcuA. To this end, we performed analytical SEC experiments with a preformed equimolar AcsA•AcuA complex in the

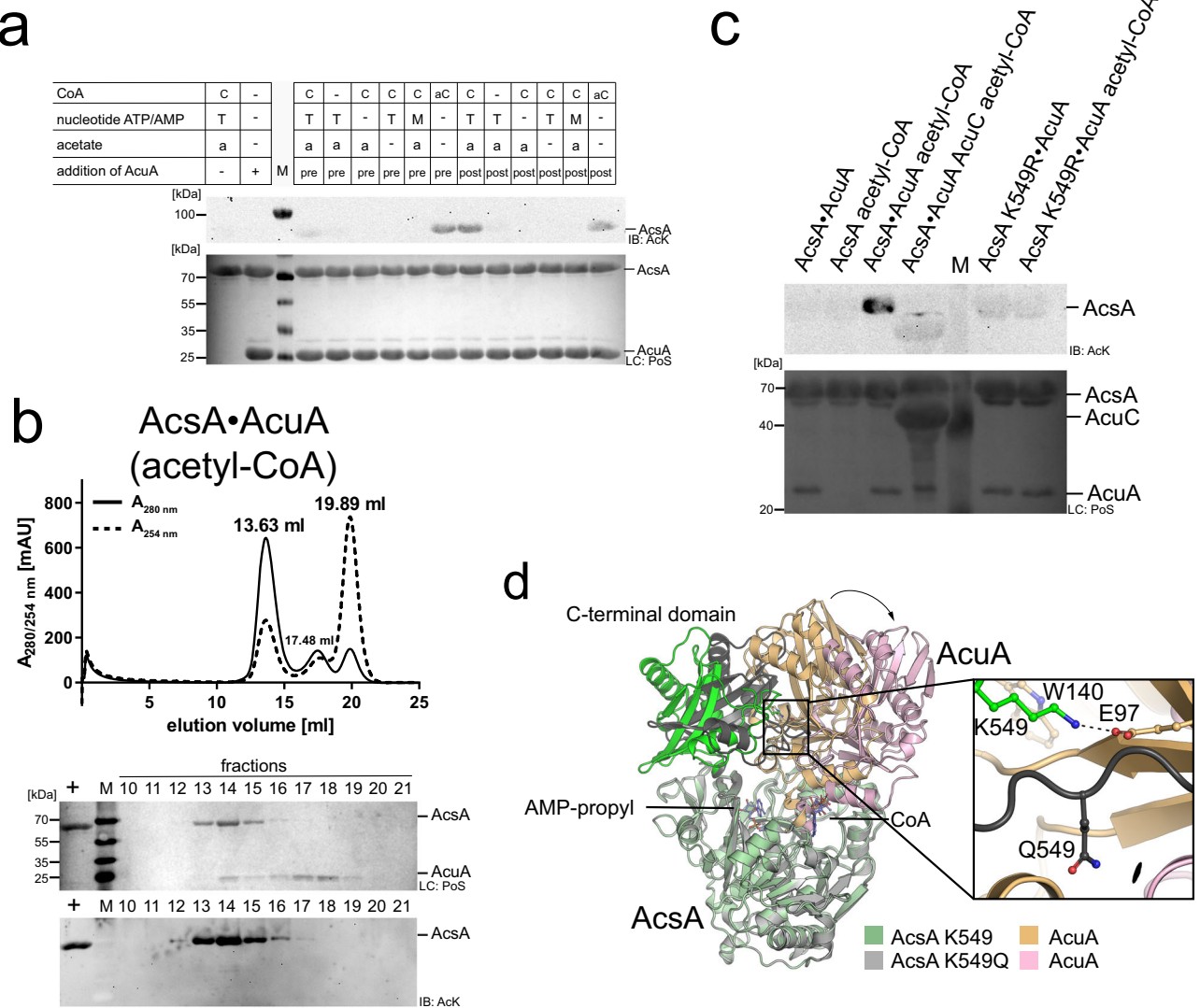

**Fig. 2 | Binding of AcuA to AcsA inactivates AcsA activity and acetylation of AcsA at K549 by AcuA results in dissociation of AcuA from AcsA. a** Binding of AcuA to AcsA inactivates AcsA-activity. The preformed AcsA•AcuA complex (pre) or AcsA alone (post) was incubated with/without CoA (C)/acetyl-CoA (aC), ATP/AMP and acetate, as indicated. Afterwards, AcuA was also added to the post samples and all samples were incubated. The acetyl-CoA generated by acetyl-CoA synthetase (AcsA) activity was detected indirectly by immunoblotting with an anti-acetyl-lysine antibody (IB: AcK) assessing AcsA K549-acetylation. The result was confirmed in at least two independent experiments. Lane M represents the protein molecular weight marker. Source data are provided as Source Data file. **b** The AcsA•AcuA complex dissociates upon acetylation of AcsA by AcuA in the presence of acetyl-CoA. SEC runs were performed with the AcsA•AcuA complex pretreated with acetyl-CoA. AcsA elutes as homodimer (13.63 ml), AcuA as monomer (17.48 ml). The peak at 19.89 ml corresponds to CoA/acetyl-CoA. Immunoblotting (IB: AcK) shows AcsA

lysine acetylation and Ponceau S-red staining (PoS) shows AcsA•AcuA dissociation. Lane + indicates the acetylated *B. subtilis* AcsA used as technical control, the M represents the protein molecular weight marker. The experiment was repeated independently three times with similar results. Source data are provided as Source Data file. **c** Acetylation of AcsA K549 is performed by AcuA and can be reversed by AcuC. AcsA or AcsA K549R (10 μM) was incubated with AcuA (2 μM) in the presence/absence of acetyl-CoA (0.5 mM) or the deacetylase AcuC (20 μM), as indicated. Samples lacking AcuC contained deacetylase inhibitor SAHA (50 μM). The lane labelled with M represents the protein molecular weight marker. The result was confirmed in at least three independent experiments. Source data are provided as Source data file. **d** AcsA K549Q alters the conformation of the AcsA•AcuA complex. AlphaFold2 structure predictions suggest conformational changes in the AcsA•A-cuA complex upon mutation of AcsA K549Q. The AcsA K549Q C-terminal domain moves towards AcuA. Q549 is not oriented towards the AcuA active site.

presence of a tenfold molar excess of acetyl-CoA. While AcsA and AcuA form a stable complex in the absence or in the presence of a tenfold molar excess of CoA or AcP (Fig. 1d; Supplementary Figs. 2c and 4c), the addition of acetyl-CoA results in dissociation of AcuA from the AcsA•AcuA complex (Fig. 2b, Supplementary Fig. 2e). The fractions of the SEC run were analysed by immunoblotting using an anti-acetyl-lysine antibody (anti-AcK-AB). The addition of acetyl-CoA to AcsA and AcuA resulted in a strong acetylation of AcsA (Fig. 2b, c). To show that K549 in AcsA is the major acetyl-group acceptor site by AcuA-catalysed acetyl-transfer, we used AcsA K549R and analysed AcsA acetylation by immunoblotting (Fig. 2c). Acetylation of AcsA K549R by AcuA is

completely abolished confirming that K549 is the major acetyl-group acceptor site in enzymatically catalysed AcsA acetylation (Fig. 2c). This also confirms that under these conditions AcsA•AcuA is not non-enzymatically acetylated at other lysine side chains to a level that can be detected by immunoblotting (Fig. 2c). We also show that AcuA-catalysed K549-acetylation on AcsA can be completely reversed by the *B. subtilis* deacetylase AcuC (Fig. 2c). This supports the molecular mechanism by which AcuA dissociates from K549-acetylated AcsA after catalysing the acetylation reaction (Fig. 2b). We also varied the molar excess of acetyl-CoA over AcsA and AcuA. Again, we observed dissociation of the AcsA•AcuA complex and strong acetylation of AcsA

suggesting that this acetyl-CoA-dependent regulation of AcsA•AcuA complex dissociation and AcsA activity is able to sense a broad range of intracellular acetyl-CoA levels (Supplementary Fig. 5a). Next, we wondered how acetylation of K549 in AcsA by AcuA results in dissociation of the AcsA•AcuA complex and leads to inhibition of AcsA activity.

**Mechanism underlying dissociation of the AcsA•AcuA complex**
To unravel the underlying mechanisms leading to the dissociation of the AcsA•AcuA complex, we first analysed the acetylation-mimetic mutant AcsA K549Q and the non-acetylated mutant AcsA K549R on its capacity to form a stable complex with AcuA by analytical SEC experiments. These show that indeed when loading equimolar concentrations of AcsA K549Q and AcuA, free AcuA eluted at 17.62 ml next to the AcsA K549Q•AcuA complex at 13.33 ml (Supplementary Fig. 6). The fact that K549Q does not completely abrogate complex formation with AcsA shows that the AcsA mutation K549Q does not perfectly mimic acetylation at K549 in AcsA. AcsA K549R•AcuA behaved as non-acetylated AcsA•AcuA forming a stable complex co-eluting in a single peak as apparent heterotetramer when analysed at equimolar concentrations by analytical SEC (Supplementary Fig. 6). As the experimental data suggest by using K549 to Q and K549 to R AcsA mutants the impact of K549-acetylation can at least partly be mimicked we performed AlphaFold2 structure predictions of these mutants and their complexes with AcuA to unravel the underlying mechanisms resulting in complex dissociation upon K549-acetylation catalysed by AcuA bound in the AcsA•AcuA complex (Fig. 2d; Supplementary Fig. 5b; Supplementary Data 1). Recent data on AlphaFold2 suggest that it is valuable to derive hypotheses although it is not perfect to predict all protein structures with high accuracy, particularly for some protein-protein interactions, protein conformational changes, side-chain conformations and ligand binding[103]. As described for wildtype AcsA•AcuA, in both structures the C-terminal domain of AcsA is reoriented to stabilize AcsA in the conformation incompatible with catalysis of both half-reactions (Fig. 2d; Supplementary Fig. 5b). Notably, we showed before that these mutations are no perfect tools to mimic the full impact of lysine acetylation at the molecular level and the real impact might be a combination of electrostatic effects, represented by the K549Q mutant, and of steric effects, represented by the K549R mutant of AcsA[1,104–108]. Superimposing the structural model of wildtype AcsA•AcuA and AcsA K549R•AcuA shows that these structures show a high degree of structural similarity with an r.m.s.d. value of 1.422 Å (Supplementary Figs. 2g and 5b). The complexes share an almost identical interface area (AcsA•AcuA: 1764.4 Å$^2$; AcsA K549R•AcuA: 1771.5 Å$^2$) suggesting a comparable stability supporting the analytical SEC experiments (Supplementary Figs. 2d and 6). AcsA K549R as wildtype K549 of AcsA is able to form several salt-bridges with conserved aspartate/glutamate residues in AcuA, i.e. D82, E85, E97, E102 and E135, as judged from the predicted AlphaFold2 AcsA•AcuA structure (Fig. 1e; Supplementary Fig. 4b; Supplementary Data 1). In contrast, AlphaFold2 structure predictions of the acetylation-mimetic mutant AcsA K549Q with AcuA suggest a conformational movement of the AcsA C-terminal domain in the complex compared to wildtype AcsA•AcuA (Fig. 2d). Q549 in AcsA K549Q is not able to form the salt-bridges with the conserved aspartate/glutamates in AcuA supporting this alteration results in a decrease of interface area from 1764.6 Å$^2$ in wildtype AcsA•AcuA to 1505.08 Å$^2$ in AcsA K549Q•AcuA (Supplementary Fig. 2d). The AlphaFold2 predicted structure of the AcsA K549Q•AcuA complex shows the C-terminal domain of AcsA K549Q in a conformational state between that observed in wildtype AcsA•AcuA complex and that observed in noncomplexed AcsA possibly representing an intermediate conformation in K549-acetylation-mediated dissociation of the AcsA•AcuA complex (Fig. 2d). According to this model, AcsA K549-acetylation would due to electrostatic quenching result in a reduction of binding affinity of AcuA to AcsA as it is unable to

form the electrostatic interactions with D82, E85, E97, E102 and/or E135. This is supported by analytical SEC experiments showing that compared to AcuA wildtype the double mutant AcuA E97Q/E135Q forms a less stable complex when run at equimolar concentrations with AcsA (Supplementary Fig. 6). This finally supports displacement of AcuA from AcsA•AcuA driven by the conformational change of the AcsA C-terminal domain adopting the AcsA adenylation-conformation (Fig. 2d; Supplementary Fig. 4b). Upon dissociation of the AcsA•AcuA complex by K549-acetylation, K549-acetylated AcsA remains inactive[36]. The molecular mechanisms how acetylation inactivates AMP-forming-CoA synthetase activity is still only incompletely understood. To this end, we next questioned how acetylation of AcsA inactivates AcsA activity.

**Mechanisms of inhibition of AcsA activity by acetylation**
As shown above, following complex formation of AcsA•AcuA, K549 of AcsA is acetylated by AcuA in the presence of acetyl-CoA resulting in subsequent displacement of AcuA from AcsA AcK549 (Fig. 2b). Acetylation at K549 of AcsA results in inhibition of AcsA catalytic activity, i.e. it is not able to generate acetyl-CoA from acetate, ATP and CoA[36]. To unravel the molecular mechanism of how K549-acetylation of AcsA results in the inhibition of AcsA activity, we initially prepared AlphaFold2 structural models for the noncomplexed mutants AcsA K549Q to mimic the electrostatic quenching effect and AcsA K549R to mimic a potential steric contribution of acetylation at K549. The structures of the noncomplexed AcsA mutants are almost identical to AcsA wildtype with overall r.m.s.d. values of 0.180 Å (Supplementary Figs. 5c and 2g). For both, AcsA K549Q and AcsA K549R, as for AcsA wildtype, the C-terminal domain is in the position that would allow AcsA to catalyse the first half-reaction of acetyl-CoA synthesis, i.e. the adenylation of acetate by ATP (Supplementary Fig. 5c). This conformation is incompatible with AcuA binding as the AcsA C-terminal domain occupies the AcuA binding surface area (Supplementary Fig. 4c). Moreover, the C-terminal domain of AcsA is in a state incompatible with CoA binding, which is essential for the thioester-forming reaction (Fig. 1c). However, K549-acetylated AcsA is not capable to catalyse the adenylation half-reaction to form acetyl-AMP and pyrophosphate from ATP and acetate. K549 is needed for binding the ribose of the bound nucleotide and for orienting the substrate for nucleophilic attack on the α-phosphate of ATP (Fig. 1c; Supplementary Fig. 5c). This is deduced from superimposing the AlphaFold2 structural model of *B. subtilis* AcsA and the structure of AMP-forming acetyl-CoA synthetase from *Salmonella enterica* crystallised with CoA and AMP-propyl (PDB: 1PG3; 1PG4; Fig. 1e)[51]. For the related enzyme phenylalanine activating gramicidin synthetase 1 (PDB: 1AMU) it is reported that K517, which is equivalent to *B. subtilis* AcsA K549, directly contacts the C-4' and C-5' oxygens of the ribose sugar of the bound AMP (or ATP) and it interacts with the carboxylate of the phenylalanine substrate (Supplementary Fig. 13c). It is therefore important for nucleotide binding and substrate activation for catalysis[70]. Acetylation of the highly conserved K549 in AcsA would abolish the formation of the hydrogen bonds with the ATP/AMP-ribose, sterically interfere with nucleotide binding and were not able to orient and activate the acetate substrate for catalysis of the first half-reaction by AcsA (Fig. 1c; Fig. Supplementary Figs. 5c and 13c). This is supported by mutational studies of firefly luciferase and PrpE showing that mutation of the conserved lysine in core motif A10 reduces the catalytic activity of the adenylation half-reaction with little effect on the second half-reaction[52,57]. We next wondered which catalytic strategies are employed by AcuA to achieve acetylation of AcsA at K549.

**AcuA acetylates AcsA by a general base catalytic mechanism**
A previously solved crystal structure of a homologue of *B. subtilis* AcuA from *Exiguobacterium sibiricum* (PDB: 2Q04) shows a bivalent calcium ion coordinated by several negatively charged glutamates and by a

histidine side chain (Supplementary Fig. 8c). The structure of *B. subtilis* AcuA predicted by AlphaFold2 shows a high degree of structural similarity to AcuA from *E. sibiricum* with an r.m.s.d of 0.764 Å (Supplementary Fig. 8c). The side chains coordinating the calcium ion are conserved in *B. subtilis* AcuA and correspond to E97, E135 and H139. Upon binding of AcuA to AcsA, K549 of AcsA inserts into the active site of AcuA coordinated by E97, E135 and H139 (Supplementary Fig. 8c). K549 of AcsA has a pK$_a$-value of 12.4 when present in the complex with AcuA indicating that it is positively charged at physiological pH as calculated with the APBS-PDB2PQR software suite[59]. The comparison of the position of AcsA K549 in the AcsA•AcuA AlphaFold2 structural model with 2Q04 shows the positively charged epsilon amino group of K549 positioned at the analogous position of the calcium ion in 2Q04 (Supplementary Fig. 8c). Notable, H138 (corresponding to H139 in *B. subtilis* AcuA) in 2Q04 can form two alternate conformations, one of which coordinates the Ca$^{2+}$-ion. These alternate conformations are not obtained by AlphaFold2 structure prediction (Supplementary Fig. 8c). Enhanced sampling molecular dynamics (MD) simulations were performed to decipher the catalytic strategies underlying the acetyltransferase (AcT) activity exerted by AcuA to achieve acetyl-CoA-dependent acetylation at K549 of AcsA[109]. Starting from the AlphaFold2 structural model of the AcsA•AcuA complex and adding CoA and acetyl-CoA, derived from the crystal structure of Ribosomal-protein-alanine N-acetyltransferase from *Brucella melitensis* solved in complex with Acetyl-CoA (PDB: 4JWP), the dynamics of ternary complexes of AcsA•AcuA•acetyl-CoA and AcsA•AcuA•CoA were sampled. The resulting structural ensembles reveal that acetyl-CoA and CoA bind in different conformations to AcuA (Fig. 3a,b). While the 4-phosphopantetheine moiety of acetyl-CoA adopted extended productive conformations approaching the AcuA catalytic active site within the AcsA•AcuA complex, conformations of CoA cluster distantly from the active site without contact to K549, in a proposed unproductive way (Fig. 3a). In contrast, the acetyl-group of acetyl-CoA is in vicinity to the epsilon amino group of the K549 of AcsA, allowing nucleophilic attack of the activated K549 side chain on the electrophilic C-atom of the acetyl-group of CoA (Fig. 3a, b). To favour a nucleophilic attack, K549 is activated by deprotonation. The MD simulations revealed that apart from AcuA E97 and E135, three more aspartate/glutamate residues, i.e. D82, E85 and E102, are transiently interacting with K549 of AcsA (Fig. 3c). E102 is totally conserved in AcuA from different bacterial species from the bacterial groups of *Chlostridia* and *Bacilli* supporting an important functional or structural role (Supplementary Fig. 9). For many Gcn5-related *N*-acetyltransferases a catalytic mechanism involving a catalytic glutamate acting as general base/acid has been reported[6,7,11]. We performed mutational studies to uncover the catalytic mechanism. Next to the glutamate side chains in the vicinity of K549, we also mutated H139 and W140 of AcuA. Histidine side chains can act as general base/acid during catalysis and the W140 is highly conserved and might therefore play a functional role (Supplementary Fig. 9). In fact, the presence of H139 within a 138-WHW-140 motif lowers the pK$_a$ of the H139 side chain to 2.7 rendering it very nucleophilic at physiological pH as calculated with the APBS-PDB2PQR software suite[59]. The acetylation of K549 in AcsA by AcuA in the presence of acetyl-CoA is a very fast reaction. Under the experimental conditions, we observed a strong signal of AcsA acetylation already after 3 min incubation time (Fig. 4a). Mutational studies suggest that E102 in AcuA acts as a general base as AcuA E102Q completely abolishes AcuA AcT activity towards AcsA, independently of the reaction time (Fig. 4a, b). Notably, the mutations of the other glutamic acid side chains in AcuA, i.e. E85A/Q, E97A, E135A/Q and the double mutants E85/E97Q and E97Q/E135Q, resulted in a stronger AcsA acetylation signal compared to AcuA wildtype suggesting that the mutations improve AcT activity of AcuA towards AcsA. From the simulations, we conclude that these charge-neutralizing mutations increase the residence time for K549 of AcsA at E102 of AcuA thereby

increasing the deprotonation efficiency of K549 (Fig. 3c). Mutation of the conserved W140 in AcuA, i.e. AcuA W140F and W140A, also affects the kinetics of AcsA acetylation. After 3 min, both AcuA W140A and W140F resulted in a strong reduction in AcuA-catalysed AcsA acetylation compared to AcuA wildtype. This suggests that W140 is important for the AcT activity, either playing a direct role in catalysis or being important for binding AcsA or acetyl-CoA. When the assay was conducted for 3 h at 37 °C, AcuA W140A still showed almost no AcsA acetylation activity while the signal observed for AcuA W140F was comparable to the one obtained with AcuA wildtype (Fig. 4a). Analytical SEC experiments show that both, AcuA W140A and AcuA W140F, form stable complexes with AcsA suggesting that W140 is important for catalysis and/or for acetyl-CoA binding (Supplementary Fig. 6). The mutant AcuA W140F is active in acetylating AcsA, albeit with a much slower reaction kinetics. This shows, firstly, the presence of an aromatic side chain being essential at this position for catalysis, as mutation to alanine results in inactive AcuA. Secondly, this reveals the tryptophan side chain is important as phenylalanine cannot completely compensate for tryptophan at this position. The main binding cluster during the simulation shows that side chain conformations of W140 cluster below the acetyl-CoA 4′-phosphopantetheine group creating van-der-Waals interactions (Fig. 3b). Moreover, the indole N-H moiety of W140 is in hydrogen bond distance to the carbonyl oxygen of the pantothenic acid moiety of acetyl-CoA (Fig. 3b). Both effects contribute to acetyl-CoA binding in the productive conformation explaining the kinetic effect observed for AcuA W140F compared to wildtype W140 as phenylalanine lacks an indole N-H group. The fact that W140A in AcuA is catalytically inactive while W140F is only kinetically impaired suggests a role of the W140 main chain in catalysis next to the postulated effect on acetyl-CoA binding (Fig. 4a). Our model shows that the conformations in which the side chain of W140 is binding to acetyl-CoA and positioned away from the active site are dominating (Fig. 3b). In these conformations, the main chain amide of W140 is positioned towards the active site. To this end, we postulate that the main chain amide nitrogen of W140 stabilizes the tetrahedral reaction intermediate by forming an oxyanion hole (Fig. 4b). Finally, the protonated E102 acts as a general acid to protonate the CoA thiolate facilitating the collapse of the tetrahedral intermediate (Fig. 4b). The calculated pK$_a$ for E102 of AcuA in the AcsA•AcuA complex is 6.02 suggesting that it can act as a proton acceptor (general base) and proton donor (general acid) during catalysis[59].

Mutation of H139A in AcuA slightly impaired AcT activity of AcuA. However, at longer incubation times AcsA acetylation was indistinguishable from the reaction with AcuA wildtype suggesting that H139 is not a residue directly involved in catalysis but plays a structural role (Fig. 4a).

As a summary, these results suggest that acetyl-CoA binds to AcuA within the AcsA•AcuA complex in a productive conformation. The totally conserved W140 of AcuA contributes to acetyl-CoA binding and formation of this productive conformation. We postulate the totally conserved E102 in AcuA acting as a general base to deprotonate and thereby activate K549 in AcsA allowing nucleophilic attack on the electrophilic C-atom of acetyl-CoA (Fig. 4b). During catalysis a tetrahedral intermediate is formed that is stabilized by the main chain amide of W140. The intermediate is resolved by E102 this time acting as general acid finally resulting in acetylation at K549 of AcsA (Fig. 4b). Notably, we also mutated E473 and V477 of AcsA and analysed how these interfere with AcT activity. We found that mutations E473A and V477A both impair the AcT activity of AcsA•AcuA. AcsA E473 forms an intramolecular interaction with R552, which in turn interacts with W138 of AcuA as judged by the apo structural model of AcsA•AcuA predicted by AlphaFold2 (Supplementary Fig. 11a, c; Supplementary Data 1). V477 is located on the same α-helix as E473 in the C-terminal domain of AcsA suggesting that mutation of V477A abolishes AcsA•AcuA AcT activity due to a structural effect (Supplementary Fig. 11c). Both mutations

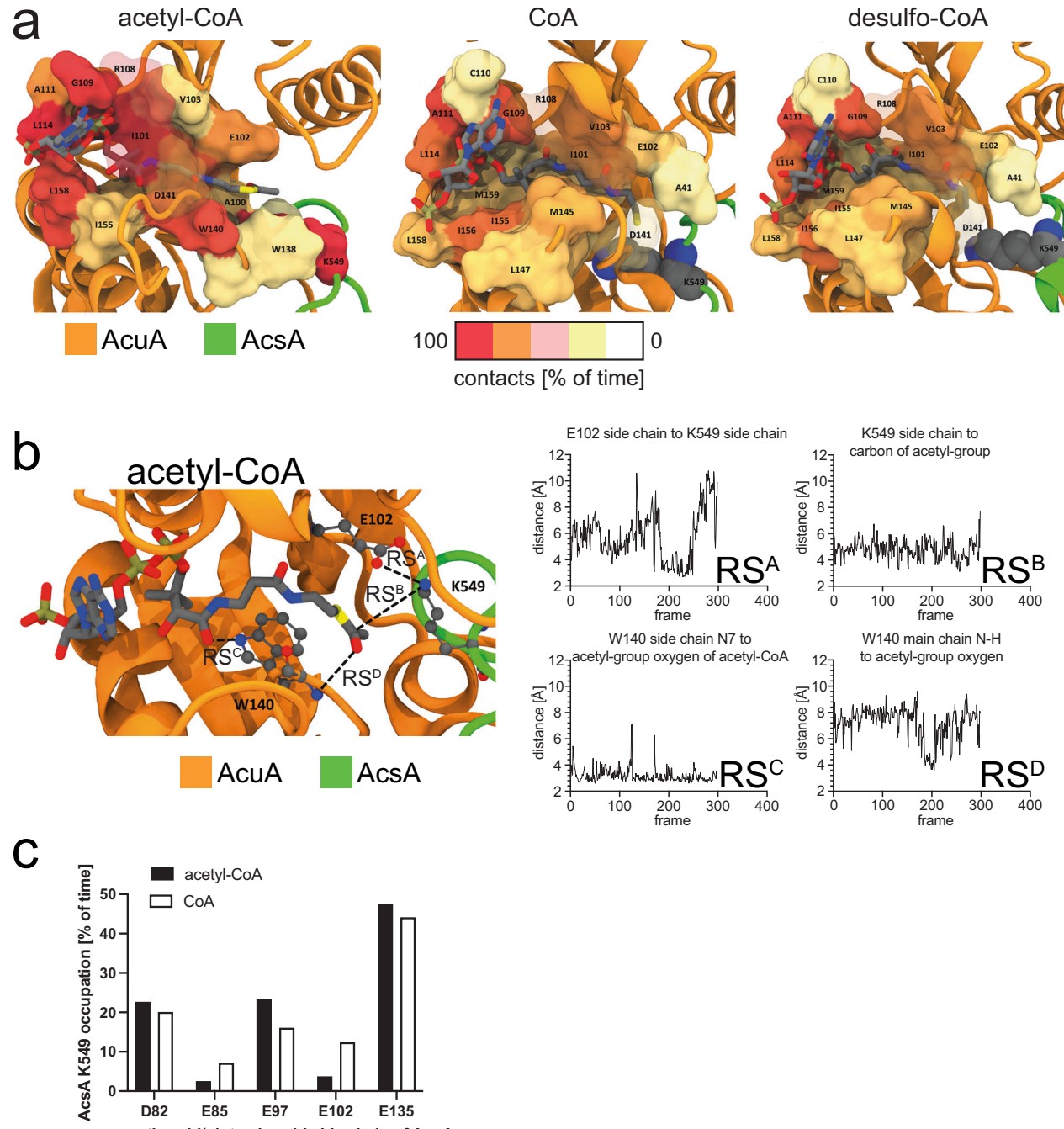

**Fig. 3 | Molecular dynamics simulations of the catalytic mechanism exerted by AcuA to acetylate AcsA. a** Main clusters for binding of acetyl-CoA, CoA and desulfo-CoA AcuA to the AcsA•AcuA complex obtained by MD simulations R2, R3, and R4, respectively (Supplementary Table 2). Binding pockets are coloured by the fraction of time with contacts between the respective CoA derivative and AcsA•AcuA and drawn as a surface for residues >20%. Acetyl-CoA binds in an extended, productive conformation and is tightly locked, while CoA and desulfo-CoA bind in an unproductive conformation with higher flexibility. Desulfo-CoA binds similar to CoA. The CoA thiol group is distant from K549 (balls) of AcsA. R108 and D141 of AcuA form a salt bridge and keep acetyl-CoA locked in, but not in the case of CoA or desulfo-CoA. Source data are provided as Source Data file. **b** Catalytic mechanism proposed for AcT activity of AcuA on K549 of AcsA based on the major cluster found in MD simulations (R2). Acetyl-CoA (sticks) binding to AcsA•AcuA. Acetyl-CoA binds in a productive conformation to AcuA in the

AcsA•AcuA complex. Side chains in AcuA E102, K549, and W140 (ball and stick) and their interaction distances are highlighted (charts RS$^A$-RS$^D$; RS: reaction step). MD simulations suggest E102 of AcuA acting as a general base/acid during catalysis (RS$^A$). The acetyl-group is positioned deeply towards K549 of AcsA enabling acetyl-group transfer (RS$^B$). W140 is important for acetyl-CoA binding. Acetyl-CoA is bound by W140 of AcuA forming stacking interactions of the aromatic side chain and the 4-phosphopantotheine moiety of acetyl-CoA (RS$^C$). A tetrahedral intermediate is formed and stabilized by the main chain amide of W140 forming an oxyanion hole (RS$^D$). **c** MD simulations revealed that K549 of AcsA forms an electrostatic network with several negatively charged residues in AcuA (D82, E85, E97, E102 and E135). The graph denotes the average time fraction of K549 being occupied by either residue via hydrogen bonding in the presence of either acetyl-CoA (R1) or CoA (R3). Source data are provided as Source Data file.

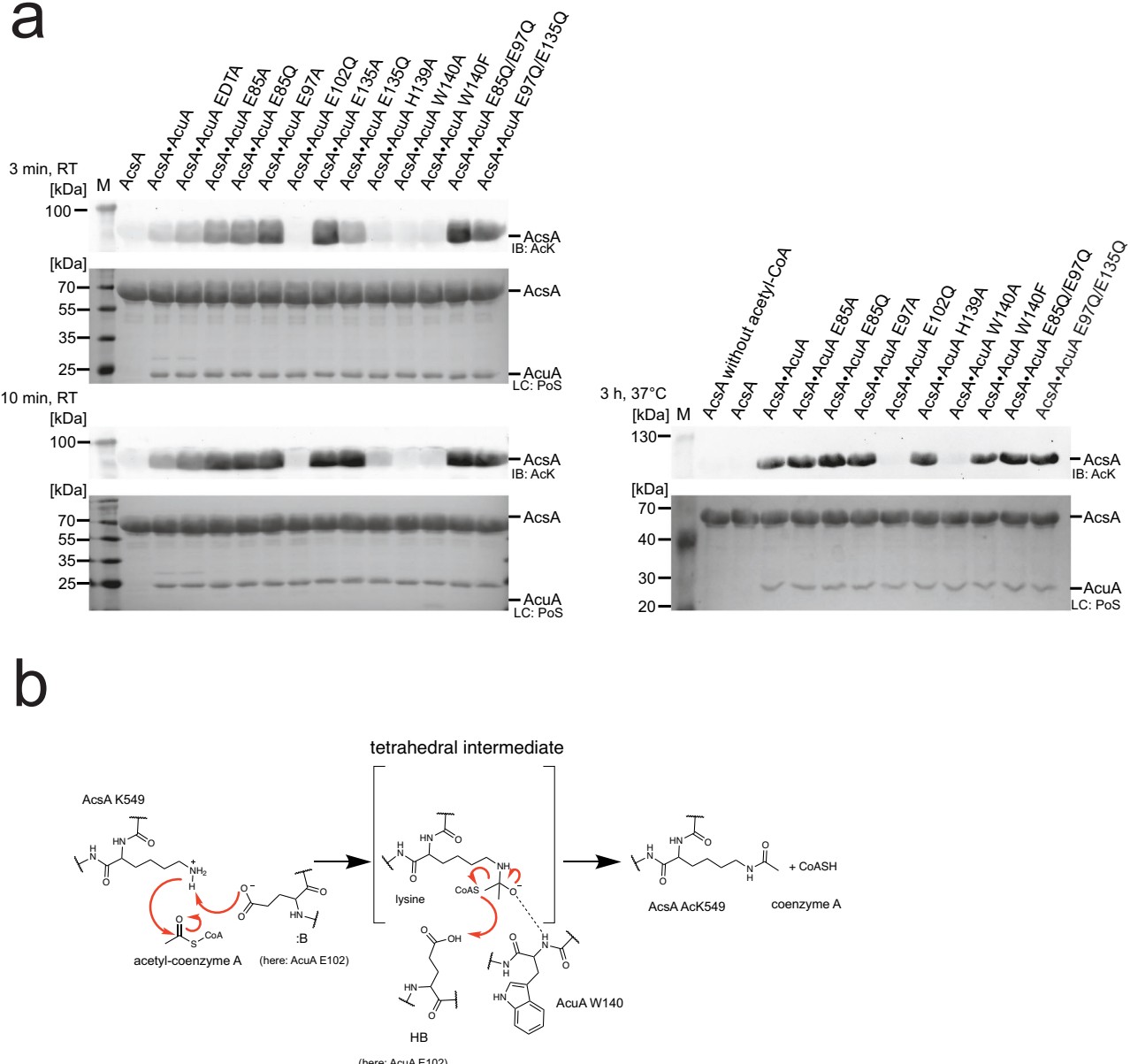

**Fig. 4 | Mutational analyses to confirm the catalytic acetyltransferase mechanism. a** Mutation of the totally conserved residues E102 and W104 of AcuA abolish AcuA AcT activity. AcuA (5 μM) was incubated with acetyl-CoA (200 μM) and AcsA (20 μM) for 3 min, 10 min at 20 °C and 3 h at 37 °C, as indicated. E102A completely abolishes AcT activity of AcuA. W140F shows a reduced AcT activity, while W140A completely abolishes the AcT activity of AcuA towards AcsA. All other mutations do not impair AcT activity. Acetylation of AcsA was assessed by immunoblotting using an anti-AcK-AB (IB: AcK). Loading control (LC) was performed by Ponceau S-red staining (LC: PoS). The lane labelled with M represents the protein molecular weight marker. The result was confirmed in at least three independent experiments. Source data are provided as Source Data file. **b** Catalytic mechanism proposed for AcT activity of AcuA on K549 of AcsA. MD simulations and mutational analyses suggest that E102 of AcuA acts as a general base/acid during catalysis. W140 is important for acetyl-CoA binding. E102 of AcuA abstracts a proton from K549 of AcsA. This enhances the nucleophilicity of K549 to attack the electrophilic carbon of the acetyl-group of acetyl-CoA. A tetrahedral intermediate is formed and stabilized by the main chain amide of W140 forming an oxyanion hole. The protonated E102 acts as catalytic acid to protonate the CoA thiolate leaving group resulting in the collapse of the intermediate to finally result in acetylated K549 and CoA.

impair the affinity of AcsA binding to AcuA as shown by analytical SEC (Supplementary Fig. 6).

Structural data on enzymes related to enzymes of the ANL-superfamily from different organisms in complexes with various ligands suggest that the enzymes can exist in various conformations, predominantly due to the high flexibility in the position of the C-terminal domain[16,21,34−36,51,58]. These data furthermore suggest that binding of ligands such as ATP, AMP, acyl-AMP, CoA or a combination thereof to the acetyl-CoA synthetase active site results in conformational changes[58]. This in turn might affect the position of K549 of AcsA,

or of other lysine side chains, affecting their accessibility for enzymatic/ non-enzymatic acetylation and/or for deacetylation by AcuC/ SrtN. To this end, we next analysed how the binding of ligands affects enzymatic and non-enzymatic acetylation of AcsA.

## AcP and CoA result in the dissociation of the AcsA•AcuA complex

To analyse the impact of ligands on AcuA-catalysed enzymatic and acetyl-CoA/AcP driven non-enzymatic acetylation of AcsA we performed immunoblottings assessing enzymatic/non-enzymatic AcsA

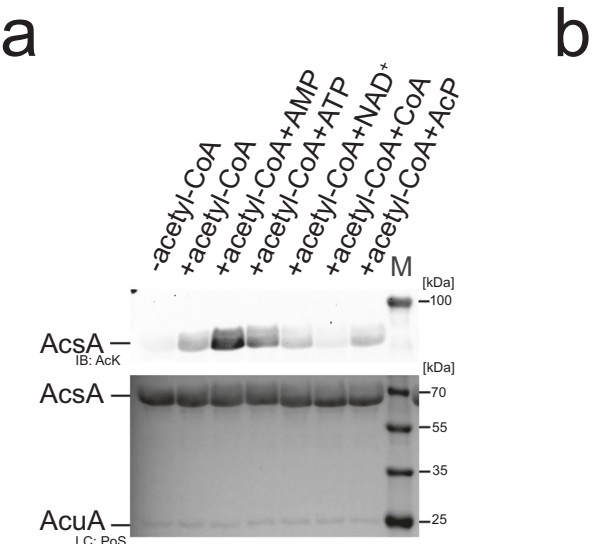

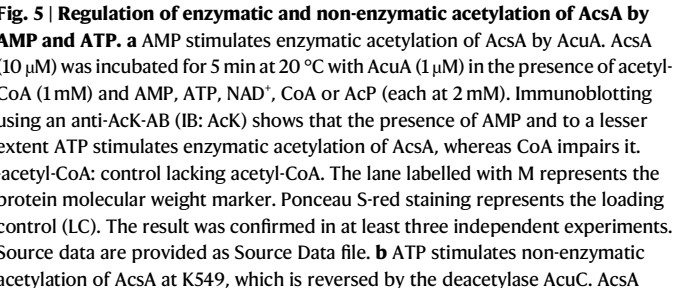

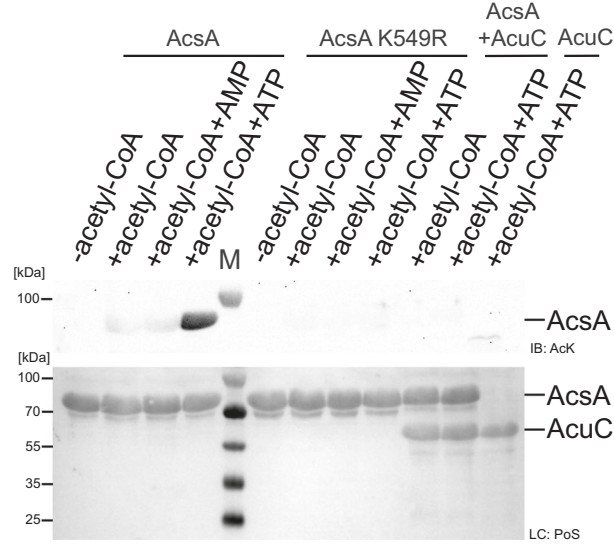

**Fig. 5 | Regulation of enzymatic and non-enzymatic acetylation of AcsA by AMP and ATP. a** AMP stimulates enzymatic acetylation of AcsA by AcuA. AcsA (10 μM) was incubated for 5 min at 20 °C with AcuA (1 μM) in the presence of acetyl-CoA (1 mM) and AMP, ATP, NAD$^+$, CoA or AcP (each at 2 mM). Immunoblotting using an anti-AcK-AB (IB: AcK) shows that the presence of AMP and to a lesser extent ATP stimulates enzymatic acetylation of AcsA, whereas CoA impairs it. -acetyl-CoA: control lacking acetyl-CoA. The lane labelled with M represents the protein molecular weight marker. Ponceau S-red staining represents the loading control (LC). The result was confirmed in at least three independent experiments. Source data are provided as Source Data file. **b** ATP stimulates non-enzymatic acetylation of AcsA at K549, which is reversed by the deacetylase AcuC. AcsA

(10 μM) was incubated for 24 h at 37 °C with acetyl-CoA (0.5 mM) and AMP/ATP (2 mM) in the presence of SAHA (50 μM)/AcuC (20 μM) as indicated. Immunoblotting using an anti-AcK-AB (IB: AcK) shows that AcsA is acetylated non-enzymatically in the presence of ATP. K549 is the major acetyl-group acceptor site for non-enzymatic acetylation in the presence of ATP as K549R shows no signal. In the presence of the deacetylase AcuC K549-acetylation is reversed. The lane labelled with M represents the protein molecular weight marker. The loading control was done by Ponceau S-red staining (LC: PoS). The result was confirmed in at least three independent experiments. Source data are provided as Source Data file.

acetylation in the presence of AMP, ATP, NAD$^+$, acetyl-CoA, CoA and AcP (Fig. 5a, b). This shows that the presence of AMP promotes enzymatic acetylation of AcsA by AcuA (Fig. 5a). Moreover, the addition of CoA impairs enzymatic acetylation (Fig. 5a). It is reported that for enzymatically catalysed lysine acetylation the acetyltransferases sense the cellular acetyl-CoA/CoA ratio rather than detecting acetyl-CoA alone, as they bind with similar affinity to acetyl-CoA and CoA[110]. In contrast, non-enzymatic acetylation of AcsA is strongly promoted by the presence of ATP and acetyl-CoA (Fig. 5b). For both, AcuA-catalysed enzymatic acetylation and non-enzymatic acetylation, K549 of AcsA is the major acetyl-group acceptor site as the mutant AcsA K549R is not acetylated in the presence of ATP/AMP and acetyl-CoA (Fig. 5b). Moreover, as described for enzymatically catalysed acetylation of AcsA, also non-enzymatic acetylation of AcsA can be reversed by the deacetylase AcuC (Figs. 2c and 5b).

The function of the AcuB protein encoded in the *acu*-operon is unknown. Structurally, it contains two N-terminal CBS (cystathionine beta synthetase)-domains and a C-terminal ACT (aspartate kinase, chorismate mutase, TyrA)-domain. Tandem pairs of CBS-domains forming a Bateman domain were reported to bind adenine/guanine nucleotides, i.e. AMP, ADP/GDP and ATP/GTP, and to NAD[+87–90]. To unravel whether the presence of ligands affects the enzymatic acetylation of AcsA indirectly by binding to AcuB, we assessed whether AcuB modulates AcuA-catalysed acetylation of AcsA depending on the presence/absence of various molecules. We also tested the impact of AcP and CoA as this would allow a regulatory feedback control mediated by AcsA acetylation (Fig. 6a). Again, we performed immunoblotting using a specific anti-AcK-AB as a readout (Fig. 6a). The presence of AcuB does not interfere with AcuA-catalysed acetylation of AcsA under these conditions not ruling out the possibility that it might play a role under different conditions not tested here. Notably, upon addition of CoA and AcP, AcsA was strongly acetylated independent of the presence of AcuB (Fig. 6a). Both, AcP and CoA, were needed for AcsA

acetylation as no signal was obtained in the sample lacking CoA (Fig. 6a). We confirmed this result by conducting analytical SEC runs with subsequent immunoblotting (Fig. 6b). Again, we pre-formed an equimolar AcsA•AcuA complex, added AcP and CoA and performed analytical SEC (Fig. 6b). Upon addition of AcP and CoA all results obtained with acetyl-CoA were recapitulated, i.e. a strong lysine acetylation of AcsA and dissociation of the AcsA•AcuA complex (Fig. 6b). Notably, the AcsA•AcuA complex formation is not affected by the presence of CoA or AcP alone as we observed a similar elution volume and calculated molecular weight compared to the experiment lacking CoA or AcP (Fig. 1d; Supplementary Figs. 2c and 4c). Moreover, neither AcsA nor AcuA was lysine-acetylated in the presence of CoA or AcP as shown by immunoblotting and the addition of CoA or AcP alone did not result in a dissociation of the AcsA•AcuA complex (Supplementary Fig. 3c). These results show that AcsA•AcuA is only able to catalyse its acetylation in the presence of AcP and CoA (Figs. 1e, 6a, and 7b). These data open the question of how the acetyl-group transfer to K549 in AcsA mechanistically occurs in the presence of AcP and CoA.

### AcuA•AcsA has an intrinsic phosphotransacetylase activity
The results shown above confirm that CoA is needed to acetylate AcsA in the presence of AcP, which opens the possibility of two reaction mechanisms for acetylation at K549 of AcsA. Firstly, AcP could be used in the presence of CoA by the AcsA•AcuA complex to directly transfer the acetyl-group from AcP to K549 of AcsA. In this scenario CoA would only play a passive role, priming the active site or orienting and binding AcP, to enable this direct transfer. Secondly, the acetyl-group is first transferred from AcP to CoA, i.e. AcsA•AcuA has an intrinsic phosphotransacetylase (Pta) activity, and the generated acetyl-CoA is subsequently used in an acetyl-transfer reaction to acetylate K549 of AcsA, i.e. by the AcT activity catalysed by AcuA on AcsA.

As a control, to exclude the possibility that we co-purified an endogenous Pta as contamination from the *E. coli* expression,

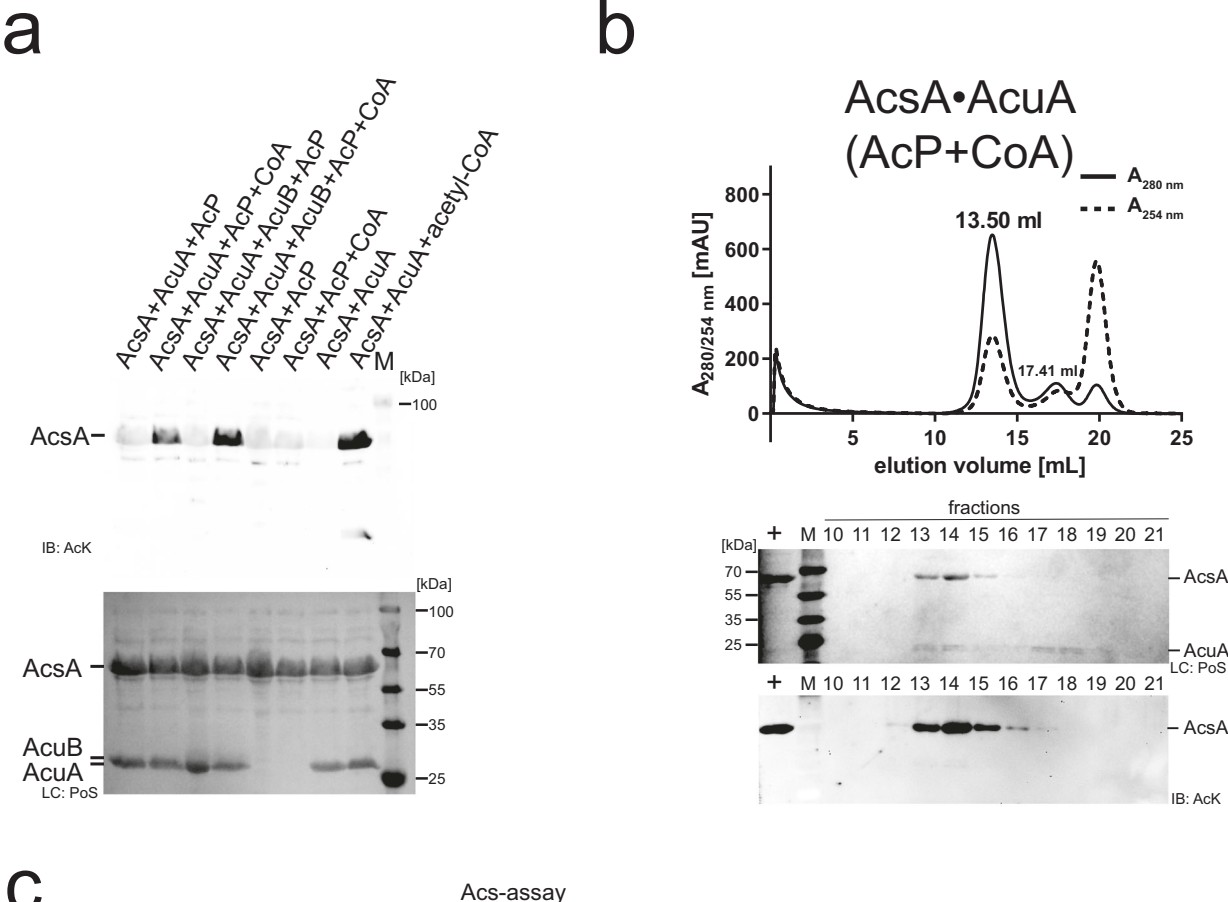

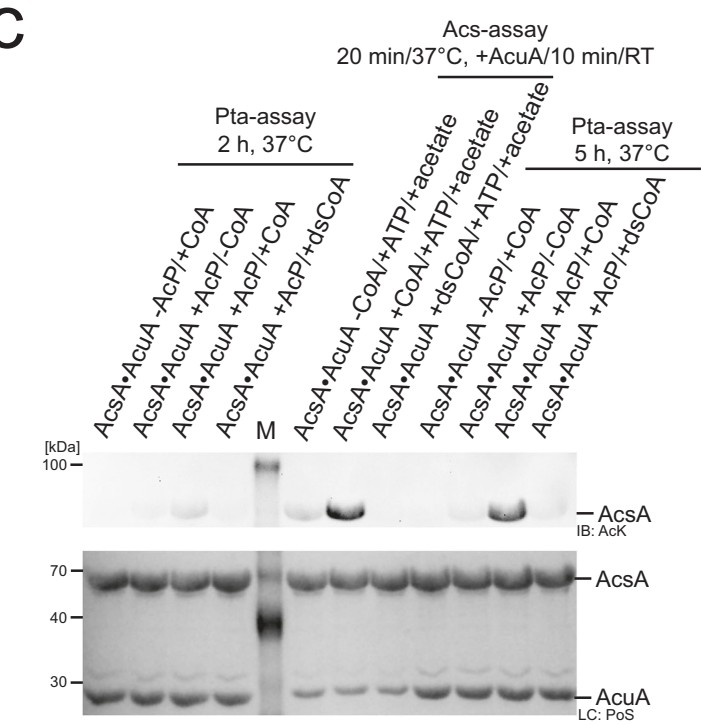

which would result in the formation of acetyl-CoA from AcP and CoA, we recombinantly expressed and purified *B. subtilis* Pta and used this in control reactions. The activity of the purified *B. subtilits* Pta was confirmed by applying a coupled-enzymatic assay (Supplementary Fig. 10a). Using this Pta we performed an analytical SEC experiment with equimolar concentrations of *B. subtilis* AcsA and *B. subtilis* Pta in the presence of tenfold molar access of

both, AcP and CoA, and analysed the eluted fractions by subsequent immunoblotting using an anti-AcK-AB (Supplementary Fig. 10b). AcsA eluted from the SEC as an apparent dimer and was not lysine-acetylated confirming that the AcsA acetylation is neither due to non-enzymatic acetylation by AcP nor by acetyl-CoA generated by the *B. subtilis* Pta activity from AcP and CoA (Supplementary Fig. 10b).

**Fig. 6 | AcuA acetylation of AcsA K549 in the presence of AcP and CoA results in AcsA•AcuA dissociation. a** AcuA (5 μM) acetylates AcsA (25 μM) in the presence of CoA and AcP independent of AcuB (25 μM). Without AcuA no acetylation of AcsA is detectable neither in the presence of AcP (2 mM) alone nor in the presence of AcP and CoA (2 mM). As a control, acetylation of AcsA is observed in the presence of acetyl-CoA (1 mM). All samples were incubated for 2 h at 37 °C. AcsA acetylation was detected by immunoblotting (IB: AcK). Lane M: molecular weight marker. Ponceau S-red staining: loading control (LC: PoS). The result was confirmed in at least three independent experiments. Source data are provided as Source Data file.
**b** Acetylation of AcsA by AcuA in the presence of AcP and CoA leads to AcsA•AcuA dissociation. Treatment of AcsA•AcuA with CoA and AcP results in acetylation of AcsA and dissociation of AcsA•AcuA (elution volume: from 13.50 ml to 17.41 ml (AcsA)) as shown by Ponceau S-red staining (PoS). Immunoblotting (IB: AcK) of SEC elution fractions shows acetylation of AcsA. Lane M: molecular weight marker. The

experiment was repeated independently three times with similar results. Source data are provided as Source Data file. **c** Desulfo-CoA cannot replace CoA in AcP-dependent acetylation of AcsA catalysed by AcuA. In the phosphotransacetylase (Pta)-assay, AcuA (10 μM) and AcsA (10 μM) were incubated with/without CoA/desulfo-CoA (dsCoA; 1 mM) and/or AcP (1 mM) for 2 h or 5 h at 37 °C. Acetylation of AcsA is only observed in samples of AcsA•AcuA containing CoA and AcP. AcsA cannot produce acetyl-CoA using desulfo-CoA (Acs-assay). AcsA (10 μM) was incubated with acetic acid (0.5 mM), ATP (0.5 mM) and CoA/dsCoA (250 μM) for 20 min at 37 °C. Afterwards, AcuA (2 μM) was added (10 min, 20 °C) to detect the generated acetyl-CoA. Detection was done by immunoblotting (IB: AcK). Ponceau S-red staining: loading control (LC: PoS). Lane M: molecular weight marker. The result was confirmed in at least two independent experiments. Source data are provided as Source Data file.

To further investigate whether AcsA•AcuA performs a Pta activity, we used desulfo-CoA, i.e. CoA in which the thiol group is replaced by a methyl group, and performed the acetylation assay of AcsA•AcuA in the presence of AcP and desulfo-CoA (Fig. 6c; Supplementary Fig. 10c). If pure binding of CoA were important to allow a direct acetyl-group transfer from AcP, a reaction should also proceed with desulfo-CoA. However, if acetyl-group transfer from AcP to the thiol group of CoA is the initial step in catalysis, the reaction is impossible with desulfo-CoA. To confirm that desulfo-CoA binds to AcuA in the AcsA•AcuA complex comparable to CoA, we conducted MD simulations to analyse the binding of desulfo-CoA in AcsA•AcuA in comparison to CoA (Fig. 3a). These simulations reveal that both, CoA and desulfo-CoA, remain and bind similarly to AcuA in the AcsA•AcuA complex entering the binding cavity in AcuA but binding in an unproductive conformation different from the binding of acetyl-CoA (Fig. 3a). As expected, AcsA is not capable to use desulfo-CoA to form acetyl-CoA with ATP and acetate (Fig. 6c, Acs-assay). Using desulfo-CoA and AcP we were not able to observe acetylation of AcsA by AcuA (Fig. 6c). We, therefore, postulate that AcsA•AcuA catalyses a Pta reaction, i.e. conversion of the mixed anhydride AcP with CoA to the thioester acetyl-CoA releasing $P_i$ (Figs. 6c and 1a). This acetyl-CoA is used subsequently by AcuA as an acetyl-group donor molecule to catalyse the acetylation of AcsA at K549. This reaction suggests that AcuA cannot directly use AcP to acetylate AcsA as the acetyl-group has to be transferred to CoA first to generate acetyl-CoA before it is subsequently transferred to K549 of AcsA (Fig. 6c). This is a report showing AcP being used for enzymatic acetylation of a protein. Next, we wondered how AcuA can catalyse both reactions, i.e. AcsA-AcT activity and Pta activity, to acetylate AcsA at K549 with the same active site.

## H139 in AcuA is needed for the Pta activity of AcsA•AcuA

As described, the MD simulations revealed differences in the binding of CoA (and desulfo-CoA) and acetyl-CoA to the AcuA active site positioning the CoA sulfhydryl-group distantly from K549 of AcsA (Fig. 3a, b). Structural sampling with the AcsA•AcuA complex and CoA/AcP suggest AcP clustering at the side chain of the conserved R108 of AcuA forming an electrostatic interaction (Fig. 7a). During the simulation of the acetyl-CoA binding to AcuA in the AcsA•AcuA complex, R108 forms a salt bridge to D141 (Fig. 3a). This salt bridge is less frequently observed in the simulations of the complexes of CoA with AcP and desulfo-CoA during the whole simulation and for the main clusters (Fig. 3a; Supplementary Fig. 10d). Notably, in the simulations with CoA and AcP, formation of this salt bridge was less frequently observed compared to the simulation with desulfo-CoA alone, suggesting that presence of the AcP competes with AcuA D141 for binding to R108 (Supplementary Fig. 10d). The interaction of R108 with AcP positions the CoA sulfhydryl group near the electrophilic carbon of AcP (Fig. 7a). Besides, the protonated CoA thiol group is frequently present near the side chain of AcuA H139 (Fig. 7a). To clarify the molecular mechanism underlying the observed AcsA•AcuA Pta activity, we performed

mutational analyses (Fig. 7b). Mutation of glutamic acid side chains in AcuA in contact distance to K549 of AcsA, i.e. AcuA E85A/Q, E97A, E135A/Q and the double mutants E85Q/E97Q, E97Q/E135Q, do not impair AcsA acetylation in the presence of AcP and CoA (Fig. 7b). Instead, we observed that mutation of E135A and E135Q in AcuA strongly increase AcsA•AcuA Pta activity (Fig. 7b, c). Comparison of AcuA proteins from different bacterial species shows that several homologous AcuA enzymes carry a substitution to glycine/alanine at the corresponding position suggesting that these substitutions intrinsically increased Pta activity compared to *B. subtilis* AcsA•AcuA (Supplementary Fig. 9). Neutralization of the negative charge at E135 might improve AcP binding in the active site diminishing electrostatic repulsion. As the Pta activity of AcsA•AcuA, i.e. generation of acetyl-CoA from AcP and CoA, precedes AcT activity, all mutants that affect AcT activity will also be defective in this Pta activity assay as it assesses K549-acetylation of AcsA by immunoblotting as readout. Along this line, we observe that AcuA E102Q abolishes AcsA acetylation in the presence of AcP and CoA as it abolishes the AcT activity of AcsA•AcuA (Fig. 7b). MD simulations suggest that E102 in AcuA does not play a role as a general base for activation of CoA as it is located distantly from the CoA thiol group (Fig. 7a). The mutations AcuA W140A/F affect the AcT activity, which occurs with a very fast kinetic. In the Pta assay with a much slower kinetic these mutations are almost as active as AcuA WT, with slightly impaired activity observed for AcuA W140A compared to W140F (Fig. 7b). This signifies that the Pta activity is not affected by mutation of W140 in AcuA and its effect on the AcT activity is rate-limiting (Fig. 7b). We observed that H139A in AcuA impairs the Pta activity of the AcsA•AcuA complex compared to AcsA•AcuA wildtype, while it shows only a mild impact on the AcT activity, most likely by indirectly affecting position of W140 (Figs. 4a and 7b). This suggests that H139A of AcuA is a mutation discriminating between AcT and Pta activity. Simulations show the CoA thiol group in close vicinity to H139 of AcuA (Fig. 7a). H139 is located in a conserved 138-WHW-140 motif. The calculated $pK_a$ value of this H139 at pH 7 is 2.7 capable of acting as a strong nucleophilic general base, deprotonating the CoA thiol group for nucleophilic attacking the electrophilic carbon of the acetyl-group in AcP[59]. We postulate that binding of AcP to the positively charged guanidino group of the conserved R108 enables efficient nucleophilic attack of the deprotonated CoA thiolate on AcP by neutralizing the negatively charged phosphate of AcP (Fig. 7a, d). Following acetyl-group transfer from AcP to CoA (Pta activity) the acetyl-CoA could convert into the productive conformation evoking the AcT activity of AcuA within AcsA•AcuA as described above. We also analysed several AcsA mutants on their impact on Pta and/or AcT activity. These results suggest that all AcsA mutations that affect the AcT activity are also impairing Pta activity (Supplementary Fig. 10b). Moreover, AcsA K549 is also the major acetyl-group acceptor site of acetylation using AcP and CoA as substrates. The acetylation is reversed by the deacetylase AcuC as described for the AcT activity (Fig. 7c). Overall, our findings show that by exerting two activities, namely an AcT activity and a Pta

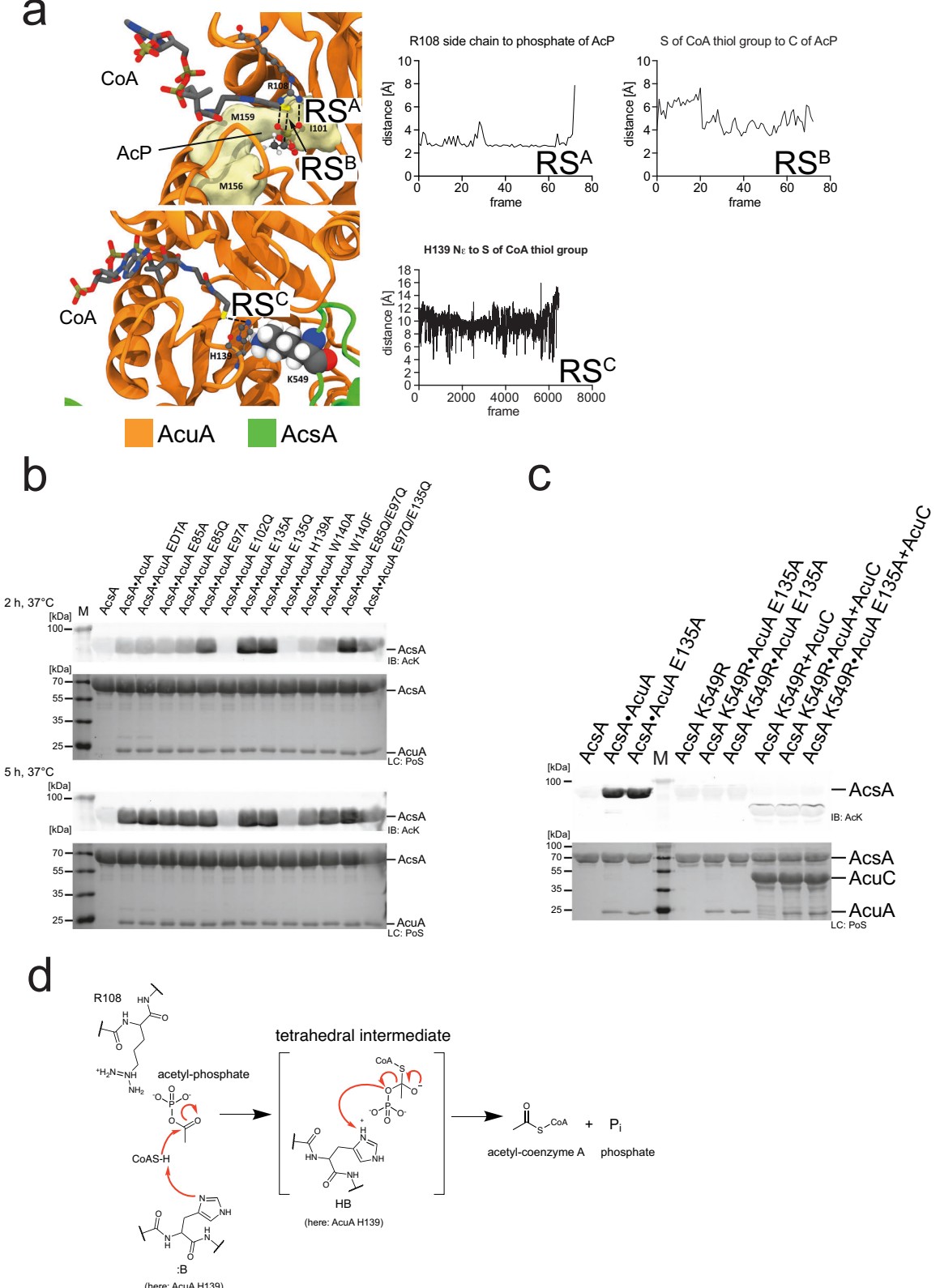

activity, AcsA•AcuA is able to sense cellular acetyl-CoA and AcP levels, and thereby adjust the AcsA activity and the generation of acetyl-CoA to the cellular metabolic state (Fig. 8).

## Discussion

Bacteria use post-translational lysine acetylation to control various cellular processes, many of which allow the sensing of the metabolic state and translating this into altered protein functionalities[2,4,83,86]. As sensors of the metabolic state, acetyltransferases use acetyl-CoA as a donor molecule for the acetylation of the ε-amino group of lysine side chains or the α-amino groups of proteins[2]. Sirtuin deacetylases use NAD⁺ as stoichiometric co-substrate to deacetylate lysine side chains[2]. The level of cellular acetyl-CoA is regulated by several processes. Acetyl-CoA is produced during β-oxidation of fatty acids, breakdown

**Fig. 7 | H139 is important for the Pta activity of AcsA•AcuA. a** Main cluster of AcP within the active site of AcuA in the AcsA•AcuA complex from MD simulation (R3). R108 binds the phosphate group of AcP (RS^A; RS: reaction step) and the acetyl-group is stabilized in a hydrophobic pocket. The thiol group of CoA is close to the acetyl-group C-atom of AcP (RS^B), priming it for transfer of the acetyl-group. MD simulations reveal H139 of AcuA coming in close contact with the thiol of CoA, allowing to abstract a proton to activate CoA (RS^C). Source data are provided as Source Data file. **b** Mutational studies reveal that H139 is essential for Pta activity. AcsA (20 μM) was incubated (2 h or 5 h; 37 °C) with AcuA/AcuA mutants (5 μM), AcP (1 mM), CoA (2 mM). EDTA (10 mM) was used to unravel a potential contribution of a metal ion on catalysis. The result was confirmed in at least three independent experiments. Source data are provided as Source Data file. **c** In the presence of AcP and CoA K549 of AcsA is acetylated within the AcsA•AcuA complex. Acetylation is increased upon mutation of E135 in AcuA and is reversed by the classical deace-tylase AcuC. AcuA (1 μM) was mixed with AcsA (10 μM) in the presence of 1 mM AcP and 2 mM CoA, incubated for 3 h at 37 °C, and then, AcuC was added for 30 min at 37 °C. Acetylation of AcsA was analysed by immunoblotting (IB: AcK). Ponceau S-red staining: loading control (LC: PoS). The result was confirmed in at least three independent experiments. Source data are provided as Source Data file. **d** Mechanism of Pta activity via H139 acting as a general base. The results suggest H139 acting as a general base to deprotonate and thereby activate the CoA thiol group for nucleophilic attack on the electrophilic C-atom of the acetyl-group of AcP. This is enhanced by R108 of AcuA neutralizing negative charges at the AcP phosphate. A tetrahedral intermediate forms being resolved by H139 acting as general acid protonating the phosphate leaving group finally resulting in the formation of acetyl-CoA and inorganic phosphate.

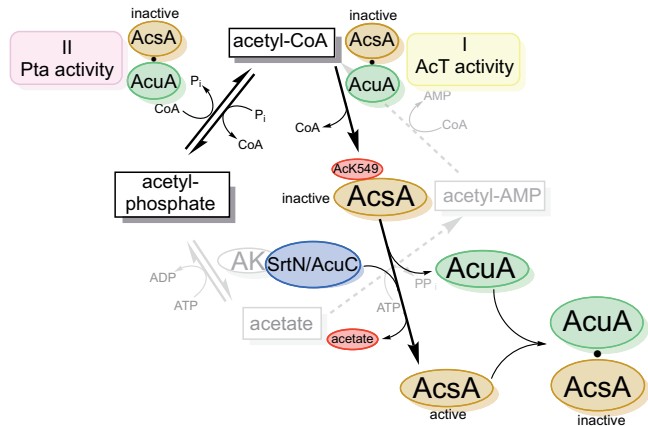

**Fig. 8 | Model for regulation of AcsA activity via acetylation by AcT and Pta activity.** AMP-forming acetyl-CoA synthetase AcsA forms a complex with the Gcn5-related acetyltransferase AcuA, inhibiting AcsA activity by stabilizing AcsA in a conformation incompetent to catalyse the first or second half-reaction of acetyl-CoA synthesis. In the presence of acetyl-CoA, AcuA acetylates AcsA at K549 in the C-terminal domain (I: AcT activity). The acetyltransferase reaction proceeds with a general base-mechanism, i.e. E102 of AcuA. K549-acetylation abolishes electrostatic interactions of K549 with several aspartic acid/glutamic acid side chains of AcuA, i.e. D82, E85, E97, E102, E135. In turn, this lowers the affinity of AcuA to AcsA enabling the AcsA C-terminal domain to displace AcuA adopting the adenylation-conformation of the first half-reaction in the apo form. K549-acetylated AcsA is inactive as it interferes with ATP/acyl-AMP binding and it is not capable of orienting the substrate for nucleophilic attack on the α-phosphate of ATP. In the presence of AcP and CoA AcuA•AcsA catalyses a phosphotransacetylase reaction (II: Pta activity) involving H139 of AcuA as general base to deprotonate the thiol of CoA attacking the AcP generating acetyl-CoA, which can be used by AcuA in a subsequent acetyltransferase reaction to acetylate K549 of AcsA. This enables to switch off acetyl-CoA synthetase activity under conditions of high levels of CoA and AcP (acetate dissimilation). Furthermore, it dissociates the AcsA•AcuA complex under conditions of a low ratio of acetyl-CoA/CoA, under which AcuA activity is low, resulting in replenishing a pool of noncomplexed, readily activatable K549-acetylated AcsA. Re-activation of K549-acetylated AcsA is accomplished by deacetylation by the classical Zn^{2+}-dependent deacetylase AcuC or a sirtuin deacetylase SrtN. Sections in light grey are shown to contextualize the regulation of AcsA activity into the metabolic pathways.

of ketogenic and glucogenic amino acids and by the activity of the pyruvate dehydrogenase (PDH) complex. Moreover, acetyl-CoA can be generated by the action of AMP-forming acetyl-CoA synthetases present in eukaryotes and bacteria[35,43–45,51,76,79,111–114]. For mammalian cytosolic and mitochondrial AMP-forming enzymes, AceS1 and AceS2, a regulation via lysine acetylation has been found[43]. The acetylation of AMP-forming acetyl-CoA synthetases is enzymatically catalysed by acetyltransferases and can be reversed by classical Zn^{2+}-dependent enzymes and by NAD^+-dependent sirtuins both, in prokaryotes and eukaryotes[14,34,35,43,55,60,115]. So far acetylation of AMP-forming acetyl-CoA

synthetases was exclusively shown to be catalysed by GCN5-related N-acetyltransferases suggesting that the observed Pta activity might be realized also in other systems[21,34,60,80,116]. The mitochondrial enzyme AceS2 is regulated by the sirtuin SIRT3, however, until recently no robust lysine acetyltransferase (KAT) was identified in mitochondria[43]. The KAT MOF was shown to be able to translocate to the mitochondrial matrix affecting mitochondrial function and integrity[117,118]. The AK/Pta pathway was not identified in mammals so far, however, AcP was shown to be present in mitochondria albeit at lower concentrations compared to bacteria[119]. Future studies will reveal whether MOF can enzymatically acetylate AceS2 and whether AceS2 has an intrinsic Pat activity. Alternatively, AceS2 might only be acetylated non-enzymatically in the mitochondrial matrix driven by the high concentrations of acetyl-CoA and the slightly basic condition in this cellular compartment[120,121]. The catalysed reaction occurs in two half-reactions. Firstly, acetyl-adenylate (acetate-AMP) is formed from acetate and ATP (adenylation). This reaction is almost irreversible by the production of pyrophosphate which is converted into two molecules of phosphate by pyrophosphatase. Secondly, the mixed anhydride acetate-AMP is converted into the acetyl-CoA by the formation of a thioester (thioester-formation)[54,113]. This reaction consumes CoA and AMP is released. For *B. subtilis* AcsA and several eukaryotic and bacterial AMP-forming acetyl-CoA synthetases it is known that the activity is inhibited by acetylation of a highly conserved lysine side chain in core motif A10[16,44,46,60,80,115,122]. In *B. subtilis*, this acetylation is catalysed enzymatically by the acetyltransferase AcuA[80]. This constitutes a negative regulatory feedback loop. AcuA activity correlates with the intracellular acetyl-CoA level resulting in inhibition of AcsA activity when cellular acetyl-CoA levels are high. In *B. subtilis*, acetylation of AcsA can be reverted by the classical Zn^{2+}-dependent deacetylase AcuC and by the NAD^+-dependent sirtuin deacetylase SrtN[34]. Under conditions of high intracellular NAD^+ levels AcsA activity and thereby production of cellular acetyl-CoA is increased due to AcsA deacetylation. Overall, the similar regulation mode of AMP-forming acetyl-CoA synthetase by post-translational acetylation and deacetylation suggests an evolutionary highly conserved mechanism[50]. Here we discovered a metabolic branch directly connecting the AK/Pta pathway and the AMP-forming acetyl-CoA synthetase pathway (Fig. 8). This allows adjusting the activity of AMP-forming acetyl-synthetase to the intracellular concentrations of acetate, acetyl-CoA/CoA and AcP. We present data showing that AcsA and AcuA directly interact and that AcP and CoA can be used by AcuA in complex with AcsA to directly acetylate AcsA at K549. Our data reveal that the reaction is dependent on the presence of AcP and CoA. We propose a model in which AcsA•AcuA uses AcP for acetylation of CoA to generate acetyl-CoA (Pta activity) that is subsequently used by AcuA to acetylate K549 and inactivate AcsA (AcT activity). Thereby, the cellular AcP and CoA levels can directly be translated into modulated AcsA activity, i.e. under conditions of high intracellular AcP and CoA, AcsA activity can be inhibited by AcuA-mediated K549-acetylation. Under conditions of high intracellular concentrations of acetate observed under conditions of

carbon overflow, acetate can be converted via the AK/Pta pathway to generate acetyl-CoA[100]. The $K_M$-values of the enzyme AK for acetate and Pta for AcP were reported to reside in the millimolar range[102]. In contrast, AMP-forming acetyl-CoA synthetase has a $K_M$-value for acetate in the micromolar range (app. 200 μM), i.e. it is used for the assimilation of acetate at lower intracellular acetate concentrations[48,100]. The intrinsic Pta activity of AMP-forming acetyl-CoA synthetase we describe here presents a shunt allowing to switch off AMP-forming acetyl-CoA synthetase activity by AcP and CoA. Acetate utilization, i.e. acetate assimilation, is predominantly catalysed by AMP-forming acetyl-CoA synthetase as AK is active only under conditions of very high, millimolar intracellular acetate concentration such as observed in carbon overflow metabolism[48,100]. This ensures that under these conditions no two pathways are active, i.e. AMP-forming acetyl-CoA synthetase and AK/Pta pathway, resulting in the generation of acetyl-CoA. Moreover, the AK/Pta pathway is reversible in contrast to AMP-forming acetyl-CoA synthetase, and plays, therefore, an important role in acetate production, i.e. acetate dissimilation, from acetyl-CoA[48,100]. Under conditions of high intracellular acetyl-CoA levels, AcP and CoA are produced by Pta from acetyl-CoA, allowing to directly modulate AMP-forming acetyl-CoA synthetase activity and ensuring that acetate is not directly used to re-generate acetyl-CoA. Importantly, the activity of bacterial GNAT KATs is not regulated by intracellular acetyl-CoA levels alone but rather by the acetyl-CoA/CoA ratio as the $K_M$-values of bacterial GNAT KATs are similar for CoA and acetyl-CoA[110,123]. We show that the presence of CoA does not affect the complex formation of AcuA and AcsA. However, we confirm that at high intracellular concentrations of CoA, the AcT activity of AcuA is impaired ultimately preventing acetylating and inhibiting AMP-forming synthetase activity. The observed Pta activity allows inhibition of AcsA under conditions of high cellular CoA and AcP levels, which disfavours acetyl-CoA-dependent lysine AcT activity. Besides, the Pta activity of AcsA•AcuA allows dissociation of the AcsA•AcuA complex under conditions of low acetyl-CoA/high CoA levels by acetylating the AMP-forming acetyl-CoA synthetase. This leads to an increasing pool of inactivated, acetylated AMP-forming acetyl-CoA synthetase readily reactivatable by deacetylation catalysed by classical $Zn^{2+}$-dependent deacetylases such as AcuC or $NAD^+$-dependent sirtuins such as SrtN[34].

Applying mutational analyses, MD simulations and structure prediction by AlphaFold2, we also shed light onto the molecular mechanism exerted by AcuA to catalyse acetyl-group transfer from acetyl-CoA to K549 of AcsA and for the Pta activity catalysed by AcsA•AcuA to generate acetyl-CoA from AcP and CoA. Our analyses on the concerted AcsA•AcuA functions applying AlphaFold2 to model the wildtype AcsA•AcuA complex and the complexes with the acetyl-K549 (AcK549) mimetic mutants AcsA K549Q and AcsA K549R revealed how K549-acetylation of AcsA results in dissociation of AcuA from the AcsA•AcuA complex. K549-acetylation of AcsA impairs binding towards AcuA. This is sufficient for the C-terminal domain of AcsA to displace AcuA while converting from the conformation observed for the AcsA•AcuA complex to the adenylation-conformation observed in AcsA alone. The inhibition of the AcsA catalytic activity by AcuA is achieved by two mechanisms. Firstly, AcuA binding to AcsA stabilizes the AcsA in an additional conformation not described before. In this conformation, AcsA is neither capable to catalyse the first half-reaction (adenylation reaction) nor to catalyse the second half-reaction (thioester-forming reaction). Secondly, K549-acetylation of AcsA by AcuA results in the dissociation of AcuA from AcsA thereby bringing AcsA into the adenylation-conformation. However, acetylation of K549 in AcsA impairs also the activity to catalyse the adenylation reaction as it interferes sterically and electrostatically with acyl-AMP/ATP-binding. Moreover, acetylated K549 is not able to orient the carboxylate of the substrate acetate for nucleophilic attack on the α-phosphate of ATP. We want to emphasize that AlphaFold2 structure predictions might

not reveal all possible conformations present for AcsA and AcsA•AcuA under physiological conditions. Several structures were available for AMP-forming enzymes in their apo states and in complexes with various ligands suggesting that these enzymes can exist in at least three states (apo forms with flexible conformations of the C-terminal domain, CoA/acyl-AMP, AMP/ATP/substrate form; Supplementary Fig. 13)[51,58,61–64,66–68,70,72,79]. The experimentally determined structure of the *Chloroflexota bacterium* AcsA (PDB: 8RPL) presented here is a structure solved in complex with acetyl-AMP, i.e. the product of the first half-reaction (Supplementary Fig. 13). It represents the thioester-forming conformation also described for acetylated/non-acetylated structures of *Salmonella enterica* Acs (PDB: 1PG3, 1PG4) in complexes with CoA and propyl-AMP[51]. In this conformation, the K517 of core motif A10 (corresponding to K549 in *B. subtilis* AcsA) is in a primed state for binding to the unknown acetyltransferase by conformational change of the C-terminal domain. In contrast to the previous report, at least in this case product formation of the adenylation reaction alone, i.e. binding of acetyl-AMP, rather than CoA binding is sufficient to stabilize the thioester-forming conformation, which is capable of subsequent CoA binding[51]. The apo structure of *Chloroflexota bacterium* AcsA (PDB: 8RPK) shows a movement of side chains involved in nucleotide binding and a peptide flip next to them, which transfers to ca. 2 Å shift of the C-terminal domain compared to the structure in complex with acetyl-AMP (Supplementary Fig. 13b). This might be explained by the observation that in the structure with acetyl-AMP, lysine and arginine side chains (K536 of core motif A8) of the C-terminal domain form interactions to the acetyl-group oxygen, the hydroxyl at C-5′ hydroxyl and/or the C-5′-phosphate of the bound acetyl-AMP phosphate (Supplementary Fig. 13c). This interaction might contribute to adapt the thioester-forming conformation in the absence of CoA (Supplementary Fig. 13a–c). These structural analyses support that the C-terminal domain of enzymes of the ANL-superfamily is highly flexible in the apo forms and can adapt further conformations stabilized by ligand binding. We show that AlphaFold2 is a powerful tool to generate initial models, however, as shown by the experimentally determined structure of acetyl-AMP bound AcsA of *Chloroflexota bacterium*, AlphaFold2 is not able to predict all conformations with various ligands or combinations thereof adapted by AcsA. We show that AcsA bound to AcuA exists in a fourth conformation (Supplementary Fig. 13e). These conformations are almost exclusively different in their orientation of the flexible C-terminal domain with respect to the N-terminal domain. Ligand binding stabilizes the C-terminal domain in the conformations to catalyse the first or second half-reaction, and without ligands, the C-terminal is flexible and can adapt to different conformations. Our structural data on *Chloroflexota bacterium* AcsA in complex with acetyl-AMP, the product of the first half-reaction, shows that in this thioester-forming conformation, K624 (corresponds to K549 of *B. subtilis* AcsA and K609 of *S. enterica* Acs) is exposed. This conformation corresponds to the structures of K524-acetylated/non-acetylated *Salmonella enterica* Acs (in complexes with CoA and propyl-AMP PDB: 1PG3, 1PG4) indicating that, after dissociation from AcuA, K549-acetylated AcsA will be capable to catalyse the thioester-forming reaction but not the adenylation reaction. In this conformation, the acetylated/non-acetylated lysine (*B. subtilis* AcsA: K549) will be accessible for a deacetylase or an acetyltransferase to be deacetylated or acetylated.

Notably, we found in the apo structure and in the acetyl-AMP-bound AcsA structure of the *Chloroflexota bacterium* a phosphorylated histidine residue, i.e. His476, in the C-terminal domain. This histidine is not conserved amongst other members of AMP-forming acetyl-CoA synthetases (Supplementary Fig. 7). In the *B. subtilis* AcsA an arginine (R402) holds the corresponding position (Supplementary Fig. 7). For succinyl-CoA synthetase of *Trypanosoma brucei* a histidine phosphorylation was described[124]. It was reported that this phosphorylation uses the γ-phosphate of an ATP or GTP as a donor for the histidine

auto-phosphorylation and it can transfer its phosphoryl-group specifically from the histidine to an ADP molecule to re-generate ATP[124]. Although the *Chloroflexota bacterium* AcsA is a monomer in the crystal structure (supported by PISA server) and analytical SEC supports the monomeric form in solution, we observed in the crystal structure that two trimers (a crystallographic as well as a non-crystallographic trimer) are formed with an app. 500 Å$^2$ buried surface area between neighbouring molecules, with the phospho-His476 close to the threefold rotation axes (Supplementary Figs. 8b, b and 13d). Whether this is relevant for a physiologically important process, such as oligomerisation, needs further validation.

This data shows that AcP can also be used in an enzymatically catalysed reaction as an acetyl-group donor molecule to acetylate a substrate protein by combining a Pta activity with an AcT activity. Evolutionary, acetyl-phosphate is a primordial molecule that can act to phosphorylate and acetylate proteins[97,98,125]. Before the development of protein catalysts, protein functions could be adjusted by non-enzymatic acetylation through the accumulation of high concentrations of reactive acetal-phosphate. The Pta activity of AcsA•AcuA, i.e. transfer of acetyl-group from AcP to CoA to form acetyl-CoA, which is used by AcuA to acetylate AcsA (AcT activity), abolishing its activity and making the reaction reversible by dissociation of AcsA•AcuA could represent an intermediate step during the development of the AcT activity allowing a direct transfer of acetyl-CoA to the lysine acceptor site. It is still important today as it allows modulating and coordinating AcsA activity also under conditions such as low intracellular acetyl-CoA and high intracellular CoA and AcP. The conservation of the active site residues suggests that this regulation of AcsA activity might be distributed in eukaryotes and bacteria. It was shown that different acetyltransferases can acetylate AMP-forming acetyl-CoA synthetases and classical deacetylases as well as NAD$^+$-dependent sirtuins can deacetylate it. Future research will elucidate whether this mechanism can be applied also to other lysine acetyltransferases and their substrate proteins or whether this is a specific mechanism only for AcuA and AcsA to regulate and adjust AcsA activity to the cellular metabolic state.

As a summary, we provide mechanistic insights into the interaction of AcuA and AcsA and unravel catalytic strategies employed for the regulation of AcsA activity by lysine acetylation catalysed by acetyl-CoA-dependent AcuA AcT activity. We discovered a yet unidentified intrinsic Pta activity for AcsA•AcuA employed to regulate AcsA activity. Regulation of the AMP-forming acetyl-CoA synthetase activity by lysine acetylation catalysed by AcuA either using acetyl-CoA (AcT) or using AcP (Pta) as acetyl-group donor molecule allows coordinating acetate dissimilation and assimilation (Fig. 8).

Moreover, this Pta activity enables the dissociation of the AcsA•AcuA complex under conditions of high cellular CoA and AcP levels allowing a re-activation of AMP-forming acetyl-CoA synthetase activity by deacetylation catalysed by the classical deacetylase AcuC and/or the sirtuin SrtN[34]. Notably, the actual AcsA acetylation state under physiological conditions is a complex interplay of several factors, i.e. concentrations of CoA/acetyl-CoA/acetate/AcP/NAD$^+$ and the expression levels of AcsA, AcuA and of the deacetylases. As a consequence, lysine acetylation AcsA activity is dynamically adjusted to diverse cellular metabolic conditions. As acyl-AMP-forming enzymes are present and conserved in all domains of life our findings might have general implications not exclusively for AMP-forming acetyl-CoA synthetases but for enzymes evolutionary conserved such as non-ribosomal peptide synthetases, luciferases and acyl-CoA synthetases. We show here our approach consisting of biochemical experiments, experimental structural analyses, AlphaFold2 structure predictions and MD simulations is powerful to develop hypotheses and to characterise processes and finally even to unravel reaction mechanisms[103].

## Methods

### Expression and purification of proteins

The *Bacillus subtilis* proteins AcuA (uniprot: P39065), and AcsA (uniprot: P39062) were expressed as N-terminally His$_6$-tagged fusion proteins in pET45b(+) in *Escherichia coli* BL21 (DE3) cells. The protein expressions were conducted in 2 L of Lysogenic Broth (LB) media. To improve the solubility of AcuB (uniprot: P39066), we co-expressed it as N-terminally His$_6$-tagged fusion proteins together with catalytically inactive deacetylase AcuC (uniport: P39067) mutant (H133A) in pRSFDeut-1, and conducted the expression in 1L TB terrific broth (TB) supplied with 0.2 mM ZnCl$_2$ The synthetic genes were codon optimized for expression in *E. coli* (Biocat GmbH, Heidelberg). For expression, cells were cultivated to an OD$_{600}$ of 0.3 (37 °C; 150 rpm). Subsequently, expression was induced by the addition of 0.2 mM of isopropyl-β-D-thiogalactopyranoside (IPTG) and was conducted for 12–16 h (18 °C; 150 rpm). Afterwards, the cells were harvested by centrifugation (4000 × g, 20 min) and resuspended in resuspension buffer (50 mM Tris/HCl pH 7.4, 50 mM KCl, 2 mM β-mercaptoethanol with 0.2 mM Pefabloc protease inhibitor). The cells were lysed by sonication and the cleared lysate (20,000 × g, 45 min) was afterwards applied to HisTrap FF (Ni$^{2+}$-precharged for AcuA and AcsA purification; Zn$^{2+}$-precharged for AcuB purification) column equilibrated with the standard buffer (50 mM Tris/HCl pH 8, 50 mM KCl, 2 mM β-mercaptoethanol). A washing step was performed with a high-salt washing buffer (standard buffer plus 450 mM NaCl and 5 mM imidazole). The elution from the Ni-NTA column was done by applying a gradient of 20–500 mM imidazole. The *B. subtilis* phosphotransacetylase (Pta; uniprot: P39646) was expressed from a pGEX-6P vector as a GST-fusion protein in *E. coli* BL21 (DE3). The expression was done as described above for the proteins from pET45b(+). For Pta, the supernatant containing the soluble fraction after cell lysis was applied to the glutathione (GSH)-affinity column equilibrated with the standard buffer. The column was washed with the washing buffer. For proteolytic cleavage of the glutathione-S-transferase (GST)-tag PreScission 3C protease was used. 2–4 mL of the digestion buffer (standard buffer plus 0.1 mg/mL PreScission 3C protease) was added to the column and incubated for 2 h on a rotator. Afterwards, the flow-through was concentrated and further purified by size exclusion chromatography (HiLoad 16/600 Superdex 75 or 200 pg; Cytiva) using standard buffer. Protein staining in SDS-PAGE gel was done with either the Thermo Scientific PageRuler Plus Prestained Protein Ladder (Thermo Fisher Scientific) or with the SERVA Triple Color Protein Standard I (Serva). The concentrated fractions were shock-frozen in liquid nitrogen and stored at −80 °C. Protein concentrations were determined by measuring the absorption at 280 nm using the proteins' extinction coefficients.

### Analytical size exclusion chromatography

To analyse the oligomeric states and the interaction of AcuA and AcsA, analytical size exclusion chromatography runs were performed on a Superdex 200 10/300 GL column (Cytiva). The SEC column was equilibrated with two column volumes of the standard buffer (50 mM Tris/HCl pH 8, 50 mM KCl, 2 mM β-mercaptoethanol). Before injecting the protein into the column, the samples were incubated for 30 min at room temperature to ensure the complex formation of AcsA and AcuA and to ensure that a potential catalytic activity of AcuA is completed. Subsequently, 100 μl of 50–100 μM of the proteins AcuA, AcsA or the AcsA•AcuA complex with/without the addition of a tenfold molar excess of CoA, acetyl-CoA and/or acetyl-phosphate were injected and the run was performed in a standard buffer. If other concentrations were used or preincubation was done, it is indicated. The protein-containing fractions were identified by following the absorption at 280 nm, $A_{280}$, to assess the absorption of the aromatic side chains. AcuA (MW: 25,483.96 g/mol) has a low molar extinction coefficient (ε) (ε: 46,870 M$^{-1}$ cm$^{-1}$) compared to AcsA (MW: 66,043.19 g/mol; ε:

$127{,}770\ \mathrm{M^{-1}\ cm^{-1}}$). Additionally, the absorption at 254 nm, $A_{254}$, was recorded by monitoring the elution of purine derivatives (CoA, acCoA) as indicated. To calculate the molecular weights of the proteins a calibration curve was done using different standard proteins (ribonuclease A (13.7 kDa,), carbonic anhydrase (29 kDa), ovalbumin (44 kDa), covalbumin (75 kDa), aldolase (158 kDa) and thyroglobulin (669 kDa)) from the low and high molecular weight calibration kit from Cytiva. A run with blue dextran was performed to calculate the void volume of the column. The partition coefficients $K_{av}$ were obtained from the elution volumes, $V_e$, the column void volume, $V_0$, and geometric column volume, $V_c$, calculated using the following equation: $K_{av} = (V_e - V_0)/(V_c - V_0)$. The $K_{av}$ values were plotted as a function of the log molecular weight (MW) and fitted using a linear equation. The resulting calibration equations are shown with the coefficient of determination, $R^2$, showing the accuracy of the fit.

## Phosphotransacetylase activity

The *B. subtilis* phosphotransacetylase (Pta) was used as a positive control. The activity of the recombinantly expressed and purified Pta was shown using a coupled-enzymatic assay developed from the manual (K-ACET kit, Megazyme). We analysed combinations of 20 μM AcuB, 20 μM AcsA, 20 μM AcuA, and 20 μM Pta for their activity to convert AcP (2 mM) and CoA (2 mM) to acetyl-CoA and inorganic phosphate, $P_i$. These reactions were performed in reaction buffer (50 mM Tris/HCl pH 8, 50 mM KCl, 2 mM β-mercaptoethanol; 2 h, 37 °C). 40 μL of the sample was mixed with 380 μL of water, 100 μL of the buffer solution (Bottle 1 of K-ACET kit, Megazyme) and 40 μL of 10 mM $NAD^+$ Solution (in $H_2O$, pH adjusted to 7.4 with $K^+/Cl^-$). After incubation for 3 min at room temperature, the absorption at 340 nm was measured. Afterwards, 4 μL of the enzyme suspension (Bottle 3 of K-ACET kit, Megazyme) was added and thoroughly mixed. The acetyl-CoA, generated by the activity of a potential Pta activity is converted by citrate synthase together with oxaloacetate to citrate and CoA. The oxaloacetate is delivered through the reaction catalysed by malate dehydrogenase by oxidation of L-malate with concomitant reduction of $NAD^+$ to $NADH + H^+$. After incubation for 15 min at room temperature, the absorption at 340 nm of the mixture was measured again. The Pta activity can be followed photometrically by an increase in absorption at 340 nm.

## Immunoblotting

For immunoblotting, the samples were separated by SDS-PAGE and afterwards, the proteins were transferred to a nitrocellulose membrane using a semi-dry immunoblotting system (1 h, 150 mA). Afterwards, the membrane was stained with Ponceau S-red solution to analyse the completeness of the transfer and as a loading control. The membrane was blocked with 5% (w/v) semi-skimmed milk PBS-T (1 h, room temperature) and afterwards incubated with the primary anti-acetyl-lysine antibody (anti-AcK-AB); abcam ab21623, 1:2000 in 5% (w/v semi-skimmed milk PBS-T; overnight, 4 °C). The membrane was washed three times with PBS-T buffer (5 min, room temperature) before the addition of the HRP-coupled secondary antibody (goat anti-rabbit-AB): abcam ab6721 (1:10,000 in 5% (w/v) semi-skimmed milk PBS-T). Detection was done by using enhanced chemiluminescence (Roth). All uncropped images are provided in the Source Data or Supplementary Information (Supplementary Figs 14–19).

## Crystallisation, data collection and refinement

Crystals for AcsA of *Chloroflexota bacterium* (uniport: A0A535FEC2) were obtained by the sitting drop vapour diffusion method in the crystallisation condition containing 25% (w/v) PEG 400, 0.1 M sodium MES pH 6,5, 0,05 M $MnCl_2$. These crystals were co-crystallised with 10 mM ATP. No additional cryoprotectant was used. Moreover, crystals were also obtained in the crystallisation condition encompassing 30% (w/v) PEG 400, 0.1 M sodium acetate pH 7,0; 0.1 M $MgCl_2$. These

crystals were co-crystallised with 10 mM coenzyme A. The crystals belonged to the hexagonal space group H32 with four molecules of *Chloroflexota bacterium* AcsA per asymmetric unit. The native data set was collected at the HZB/DESY Hamburg/Germany at 100K on beamline P13 at a wavelength of 0.9735 Å using an EIGER 16M detector. The oscillation range was 0.1° and 3600 frames were collected. The programme XDS[126] was used for indexing and integration. Scaling and merging were performed with Aimless 0.7.4[126,127]. Initial phases were determined using the programme Phaser within the CCP4 programme suite and the predicted AlphaFold2 structural model as the search model[128,129]. Refinement was done using the programme REFMAC5[130,131]. Restraints for the acetyl-AMP and coenzyme A were obtained by the programme AceDRG[132]. Coot was used for model building into the $2F_o - F_c$ and $F_o - F_c$ electron density maps in iterative rounds of refinement with REFMAC5[131,133,134]. Finally, all residues are in or close to the allowed regions of the Ramachandran plot except for Ser454 as judged by the programme MolProbity[135,136]. All structure figures were made with CCP4mg 2.11.0, PyMOL 3.0 and PyMOL 2.5.4[137,138]. Data collection and refinement statistics are given in Supplementary Table 1. $R_{work}$ is calculated as follows: $R_{work} = \sum |F_o - F_c| / \sum F_o$. $F_o$ and $F_c$ are the observed and calculated structure factor amplitudes, respectively. $R_{free}$ is calculated as $R_{work}$ using the test set reflections.

## Structure prediction using AlphaFold2

To analyse the impact of AcsA acetylation on AcsA structure, and oligomerisation with and without AcuA, AlphaFold2 version 2.3.1 was used[139–141]. In brief, the sequences of AcuA and AcsA were supplied according to the prediction, i.e. AcsA•AcuA heterodimer or AcsA homodimer. AlphaFold2 was run in monomer mode for monomers and in multimer mode for complex prediction. The multiple sequence alignments were performed using Jackhmmer on Uniref and MGnify, in addition to hhsearch (PDB70), hhblits (BFD) and Kalign. In multimer mode, PDB70 was replaced with PDB_seqres, hhbuild replaced Kalign unless some sequences were identical and another step using Jackhmmer with Uniprot was added. For monomers, 5 models were generated and 10 for complexes. Monomer models were scored using pLDDT (predicted local distance difference test) and multimers using iptm (interface ptm)+ptm (predicted TM) score. The model with the highest overall pLDDT or iptm+ptm score was used for further analyses.

## Molecular dynamics (MD) simulations

To check the stability of the predicted AlphaFold2 models, as well as to sample binding configuration for acetyl-CoA, CoA with AcP, and desulfo-CoA, the TIGER2h$_{PE}$ replica exchange algorithm was employed[142] (Supplementary Table 2). Simulations were implemented with NAMD 2.14[143] and common simulation settings[144], including the TIP3P water model and hydrogen mass repartitioning on the solute[145] (HMR) for an increased timestep of 4 fs. Systems were built using CHARMM-GUI[146] and the CHARM36 forcefield, while parameters for acetyl-CoA, CoA and desulfo-CoA with total charges of −4 and AcP with a total charge of −2 were obtained through CGENFF[147]. System charges were neutralized by appropriate numbers of sodium or chloride ions. CoA derivatives were initially placed in analogy to PDB: 4JWP, while AcP was placed outside a visible entrance passage between AcuA and AcsA, close to D141 and restrained within 15 Å by flat-bottom distance restraints between the centres of AcP and CoA and a force constant of 1 kcal/mol/Å². The box size was set to 93 × 93 × 93 Å. The AcuA and AcsA structures were restrained to 8 Å by r.m.s.d. flat-bottom restraints and a force constant of 50 kcal/mol/Å² and this was to limit the accessible conformation space of both proteins and focus the sampling on the interactions in the active site of AcuA. All TIGER2h$_{PE}$ simulations used eight replicas. The sampling and cooling phases were set to 16 ps and 8 ps, respectively. The temperature ladder spanned from 310 K up to 450 K. Convergence of the conformational ensemble

is demonstrated in the supporting information (Supplementary Fig. 14).

Interactions were analysed with an in-house code that counts atomistic contacts within a given distance between selected system components and was used to characterise the binding pockets for the CoA derivatives and AcP, or to identify and understand important interaction networks. To identify and extract specific binding configurations of CoA derivatives from the resulting state ensemble of the TIGER2h$_{PE}$ simulations, structures were first filtered by contacts below 3.3 Å between E102 and W140 to the respective CoA derivative using VMD 1.9.4[148]. In case of Acetyl-CoA, also for contacts with K549 of AcsA. Subsequently, five r.m.s.d. clusters were searched for the CoA derivative together with W140 and E102 of AcuA with a cutoff of 3 Å. Similarly, to obtain binding positions for AcP, states were first filtered by contacts below 3.3 between AcP and CoA. Afterwards, five r.m.s.d. clusters were searched for CoA together with AcP and a cutoff of 5 Å. A major cluster is commonly identified, if it covers a significantly larger fraction of states than the subsequent clusters and is referred to as the main cluster (Supplementary Table 3).

### Acetyl-CoA synthetase (Acs)-assay

To determine the activity of AMP-forming acetyl-CoA synthetase activity of AcsA or of the AcsA•AcuA complex, AcsA/AcsA•AcuA (Fig. 2a: 10 μM) was incubated with CoA (Fig. 2a: 250 μM), ATP/AMP (Fig. 2a: 500 μM) and acetate (Fig. 2a: 500 μM) as indicated. Following incubation for 30 min, 37 °C (Fig. 2a). Subsequently, as readout to detect the formed acetyl-CoA by AcsA, AcuA was added and the sample was incubated for another 20 min, 37 °C (sample: post). To assess if binding of AcuA to AcsA affects AcsA activity, AcsA (Fig. 2a: 10 μM) was preincubated with AcuA (Fig. 2a: 20 μM) for 15 min at 20 °C (Fig. 2a). All assays were conducted in assay buffer (50 mM Tris/HCl pH 8, 50 mM KCl, 2 mM β-mercaptoethanol). As a readout, an immunoblot with an anti-acetyl-lysine antibody (anti-AcK-AB) was performed. Ponceau S-red staining was done as a loading control.

### Data analysis and visualization

Raw data from most experiments was processed using Microsoft Excel 2011. Data was visualized and statistically analysed in GraphPad Prism 5. Fitting of data was also performed in GraphPad Prism 5. SnapGene Viewer 5.1.4.1 was employed for DNA sequence handling and generation of plasmid maps (SnapGene software from Insightful Science; available at snapgene.com). PyMOL 2.5.4 was used to generate visual representations of protein and DNA structures[149]. Adobe Photoshop 25.7.0 and Adobe Illustrator 28.5 were used to create figures.

### Statistics and Reproducibility

All assays were performed in independent replicates as indicated resulting in similar results. For bar graphs, the standard deviations (s.d.) and mean values were depicted. No statistical method was used to predetermine the sample size. No data were excluded from the analyses. Unpaired, two-tailed student's t-tests were performed to assess statistical significance with significance levels as indicated.

### Reporting summary

Further information on research design is available in the Nature Portfolio Reporting Summary linked to this article.

## Data availability

The X-ray structures of *Chloroflexota bacterium* AcsA in the apo state and of the structure in complex with acetyl-AMP including the structure factors and coordinates generated in this study have been deposited in the PDB database (http://www.rcsb.org) under accession codes 8RPK and 8RPL, respectively. The structural data for PDB 1LCI, 1AMU, 1PG4, 1PG3, 4JWP and 2Q04 are deposited in the PDB database. All AlphaFold2 structure predictions, including the coordinates and

the text files either containing the pLDDT (predicted local distance difference test)-scores for monomers or the iptm (interface ptm)+ptm (predicted TM) score for multimers are published in Supplementary Data 1. Source data underlying the findings of this study are provided with Source Data file and in Supplementary Figs. 15–20. Source data are provided with this paper.

## Code availability

For MD simulations the TIGER2h$_{PE}$ replica exchange algorithm was employed. The code can be found on Github and is available under the MIT license (https://github.com/SLx64/TIGER2hs). The complete structural ensembles resulting from all TIGER2h$_{PE}$ simulations in this study are available from Zenodo (https://doi.org/10.5281/zenodo.11565611)[109].

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

## Acknowledgements
This work was supported by German Research Foundation grant No. LA2984-6/1 and LA2984-8/1 (DFG, Deutsche Forschungsgemeinschaft). We also thank HZB/BESSY and EMBL/DESY for support in data collection and crystal testing.

## Author contributions
C.Q. performed most biochemical experiments and analysed the data. L.G.G., S.S., B.G., B.D., and L.B. contributed to biochemical experiments, N.G. performed molecular dynamics simulation studies and analysed the data. Mi.D. supervised the molecular dynamics simulation experiments. K.S., M.J., and L.B. conducted crystal screenings. G.P. conducted crystallographic data collection, K.S. and G.P. solved the crystal structures. Ma.D., R.S. and S.K. implemented AlphaFold2 at Greifswald university computing centre and performed AlphaFold2 structure predictions. U.B. supervised structure predictions. M.L. initiated, designed and supervised the study, contributed solving the structure and analysed the data. M.L., C.Q. and S.S. wrote the manuscript. All authors contributed to data analysis and gave comments on writing the manuscript.

## Funding

## Competing interests
The authors declare no competing interests.
