## [Peer Review File · Nature Communications]

Acetyl-CoA synthetase is enzymatically regulated by lysine acetylation using acetyl-CoA or acetyl-phosphate as donor moleculeEditorial Note: This manuscript has been previously reviewed at another journal that is not operating a transparent peer review scheme. This document only contains reviewer comments and rebuttal letters for versions considered at *Nature Communications*.

REVIEWER COMMENTS

Reviewer #1 (Remarks to the Author):

Lammers and coworkers have satisfactorily addressed my concerns. The following relatively minor issues should be addressed before publication in Nature Comm.

1. The introduction still has to be extensively edited for grammar, particularly in the new sections that were added.
2. Lines 94-106, the text that describes the structure and conserved structure, domains and sequences of AMP-forming acetyl-CoA-synthetases is still very difficult to follow and the several additional supplemental figures that are now provided don't really help simplify things since three supplemental files (two with multiple panels) are referenced. Can the authors just provide one schematic that illustrates the domains and sequences that are referenced?
3. Figure 2C, the lane labeled 'M' does not appear to contain molecular weight markers as indicated in the legend as only one protein seems to light up.

Reviewer #2 (Remarks to the Author):

AcsA is a key enzyme in acetate metabolism by forming acetyl-CoA from acetate and ATP. It has been reported that this class of enzymes is regulated by the acetylation of a conserved lysine residue that coordinates the ATP ribose. The current manuscript introduces a new regulatory mechanism for *B. subtilis* AcsA that results from the formation of a catalytically inactive complex of AcsA and AcuA constituting the dedicated acetyltransferase modifying the lysine residue (K549). By combining traditional methods of biochemistry, structural determination and modern structural predictions, the authors gained deep mechanistic insights into the catalytic mechanism and regulation of AcsA. The authors observed that acetylation of AcsA-K549 results in dissociation of the AcsA-Acu complex. Surprisingly, the authors uncovered an unexpected enzymatic activity of the AcsA-AcuA complex, which can catalyze the formation of acetyl-CoA from CoA and acetyl-phosphate. This pathway provides AcuA with the acetyl-CoA required for acetylation of AcsA-K549. Acetate-Metabolism is a key pathway in many organisms, making the topic interesting for a broad readership. The findings of the regulatory mechanism are exciting, and the authors provide detailed insights into the molecular mechanisms governing these processes. The manuscript is written in a clear and comprehensible way.

However, there are issues that need to be addressed:

1.) There are mistakes in the illustrations of reaction mechanisms that result from diversion from chemical nomenclature: The 'curved arrows' illustrate the rearrangement of electrons (breaking and forming of chemical bonds) and not the migration of protons or atoms.

Figure 3E: Protonation of the tetrahedral intermediate is illustrated wrong: The arrow should start at the S-C bond of the intermediate and the arrow head should point onto the proton of the glutamic acid.

Figure 5D left panel: Protonation of the His imidazole should be illustrated with an arrow starting on the lone pair of the imidazole nitrogen. The arrow head should point onto the CoA hydrogen, not vice versa. In addition, the fission of the pi bond in the acetate should be illustrated with an arrow starting on the double bond and pointing onto the carbonyl oxygen.

Figure 5D central panel, a similar issue as in Figure 3E: Protonation of the released phosphate should be illustrated with an arrow starting on the O-C bond of the tetrahedral intermediate and point onto the hydrogen of the imidazolium.

Please ensure that drawings of reaction mechanisms adhere to conventions for writing chemical mechanisms.

2.) The authors mention the potential impact of non-enzymatic acetylation of lysine N epsilon amines with ac(et)yl-CoA. Given the potentially higher electrophilicity of acetyl-phosphate (when compared to acetyl-CoA) and the higher nucleophilicity of the CoA thiol when compared to the lysine amine, it appears likely that acetyl-CoA will also be formed non-enzymatically. Was this side reaction not observed at all?

3.) There were a few odd phrases in the manuscript, e.g.: '!... are almost exclusively different' (line 688).

1 **Point-by-point response to the reviewers' comments**

2
3
4 **Reviewer #1 (Remarks to the Author):**

5
6 Lammers and coworkers have satisfactorily addressed my concerns. The following relatively minor issues should
7 be addressed before publication in Nature Comm.
8

9 **Response:**

10 We thank reviewer 1 for the valuable feedback and for thoroughly reading our manuscript. We think the manuscript
11 did strongly improve due to these comments. Find below our answers to the few minor points of criticism raised by
12 reviewer 1.
13

14 **Point 1:**

15 The introduction still has to be extensively edited for grammar, particularly in the new sections that were added.
16

17 Response:

18 We extensively worked on the introduction to improve the grammar.
19
20

21 **Point 2:**

22 Lines 94-106, the text that describes the structure and conserved structure, domains and sequences of AMP-
23 forming acetyl-CoA-synthetases is still very difficult to follow and the several additional supplemental figures that
24 are now provided don't really help simplify things since three supplemental files (two with multiple panels) are
25 referenced. Can the authors just provide one schematic that illustrates the domains and sequences that are
26 referenced?
27

28 Response:

29 We agree with reviewer 1 and provide another figure, i.e. Supplementary Fig. 1, illustrating the domains of AMP-
30 forming acetyl-CoA synthetase AcsA from *B. subtilis* including the sequences referenced in the text.
31

32 **Point 3:**

33 Figure 2C, the lane labeled 'M' does not appear to contain molecular weight markers as indicated in the legend as
34 only one protein seems to light up.
35

36 Response:

37 The lane labeled "M" does contain the molecular weight marker. The figure shows the immunoblot and the Ponceau
38 S-red staining of the membrane after blotting as loading control. In Figure 2C only a closeup is shown. We provide
39 the uncropped images of the immunoblot and the Ponceau S-red staining in the Source Data showing more bands
40 in the lane. It is a Serva Prestained marker.
41

42 **Reviewer #2 (Remarks to the Author):**

43
44 AcsA is a key enzyme in acetate metabolism by forming acetyl-CoA from acetate and ATP. It has been reported
45 that this class of enzymes is regulated by the acetylation of a conserved lysine residue that coordinates the ATP
46 ribose. The current manuscript introduces a new regulatory mechanism for *B. subtilis* AcsA that results from the
47 formation of a catalytically inactive complex of AcsA and AcuA constituting the dedicated acetyltransferase
48 modifying the lysine residue (K549). By combining traditional methods of biochemistry, structural determination and
49 modern structural predictions, the authors gained deep mechanistic insights into the catalytic mechanism and
50 regulation of AcsA. The authors observed that acetylation of AcsA-K549 results in dissociation of the AcsA-Acu
51 complex. Surprisingly, the authors uncovered an unexpected enzymatic activity of the AcsA-AcuA complex, which
52 can catalyze the formation of acetyl-CoA from CoA and acetyl-phosphate. This pathway provides AcuA with the
53 acetyl-CoA required for acetylation of AcsA-K549.

54 Acetate-Metabolism is a key pathway in many organisms, making the topic interesting for a broad readership. The
55 findings of the regulatory mechanism are exciting, and the authors provide detailed insights into the molecular
56 mechanisms governing these processes. The manuscript is written in a clear and comprehensible way.

57
58
59
60
61
62
63
64
65
66
67
68
69
70
71
72
73
74
75
76
77
78
79
80
81
82
83
84
85
86
87
88
89
90
91
92
93
94
95
96
97
98
99
100
101
102
103
104
105
106
107
108
109
110
111
112

Response:

We again thank reviewer 2 for the constructive feedback which strongly improves the manuscript. We carefully worked on the points of criticism raised by reviewer 2 as summarized in the following section.

Point 1:

There are mistakes in the illustrations of reaction mechanisms that result from diversion from chemical nomenclature: The 'curved arrows' illustrate the rearrangement of electrons (breaking and forming of chemical bonds) and not the migration of protons or atoms.

Figure 3E: Protonation of the tetrahedral intermediate is illustrated wrong: The arrow should start at the S-C bond of the intermediate and the arrow head should point onto the proton of the glutamic acid.

Figure 5D left panel: Protonation of the His imidazole should be illustrated with an arrow starting on the lone pair of the imidazole nitrogen. The arrow head should point onto the CoA hydrogen, not vice versa. In addition, the fission of the pi bond in the acetate should be illustrated with an arrow starting on the double bond and pointing onto the carbonyl oxygen.

Figure 5D central panel, a similar issue as in Figure 3E: Protonation of the released phosphate should be illustrated with an arrow starting on the O-C bond of the tetrahedral intermediate and point onto the hydrogen of the imidazolium.

Please ensure that drawings of reaction mechanisms adhere to conventions for writing chemical mechanisms.

Response:

We thank reviewer 2 for this comment and we totally agree with reviewer 2. The arrows in reaction mechanisms should always indicate rearrangement of electrons rather than relocation of atoms/groups of atoms. To this end, we without exception altered the figures showing the reaction mechanisms as suggested by reviewer 2.

Point 2:

The authors mention the potential impact of non-enzymatic acetylation of lysine N epsilon amines with ac(et)yl-CoA. Given the potentially higher electrophilicity of acetyl-phosphate (when compared to acetyl-CoA) and the higher nucleophilicity of the CoA thiol when compared to the lysine amine, it appears likely that acetyl-CoA will also be formed non-enzymatically. Was this side reaction not observed at all?

Response:

As already stated and explained in the response to the first revision, we did not detect non-enzymatic formation of acetyl-CoA from acetyl-phosphate and CoA. This is likely due to several reasons as stated in the point-by-point response of the first revision:

"Firstly, we did perform an experiment excluding the possibility AcP and CoA chemically forming acetyl-CoA. In Supp. Fig. 8A we show with AcP and CoA in absence of AcuA and/or AcsA no acetyl-CoA is detectable.

Secondly, the pK_a value of CoA at physiological pH is reported to be quite high (pK_a app. 9.8) (Reference: Keire D.A., Robert J.M. and Rabenstein D.L. (1992)).

Thirdly, we identified mutations in AcuA allowing to discriminate between acetyltransferase and phosphotransacetylase activity."

The bacterial cell has a metabolic pathway consisting of acetate-kinase and phosphotransacetylase to catalyze the reaction suggested by reviewer 2, i.e. formation of acetyl-phosphate from acetate and ATP and subsequent generation of acetyl-CoA from acetyl-phosphate and CoA releasing P_i. The fact phosphotransacetylase being necessary to form acetyl-CoA from acetyl-phosphate and CoA suggests the reaction being kinetically unfavorable even if thermodynamically favorable, i.e. the reaction proceeds very slowly if not enzymatically catalyzed. Even when using high concentrations of CoA and acetyl-phosphate in the millimolar range as well as long incubation times for several hours at 37°C we do not observe formation of acetyl-CoA to detectable level. If acetyl-CoA were generated we would detect it by acetylation of AcsA catalyzed by AcuA but this was not observed here.

Point 3:

There were a few odd phrases in the manuscript, e.g.: '... are almost exclusively different' (line 688).

113 Response:

114 We worked on the language and optimized the language.

115

116 Line 688:

117 "These conformations differ only in their orientation of the flexible C-terminal domain with respect to the N-terminal domain."

REVIEWERS' COMMENTS

Reviewer #2 (Remarks to the Author):

The revisions provided in the latest version have further improved the manuscript. From my perspective, no further revisions are needed.